# COVIDisgust: Language processing through the lens of partisanship

**Veranika Puhacheuskaya** *, **Isabell Hubert Lyall**, **Juhani Järvikivi**

Department of Linguistics, University of Alberta, Edmonton, Alberta, Canada

* puhacheu@ualberta.ca

**Data Availability Statement:** All the stimuli, raw data, and scripts used in this experiment are available on Open Science Framework at https://osf.io/5ep9g/.

**Funding:** This research was supported by a Social Sciences and Humanities Research Council of

## Abstract

Disgust is an aversive reaction protecting an organism from disease. People differ in how prone they are to experiencing it, and this fluctuates depending on how safe the environment is. Previous research has shown that the recognition and processing of disgusting words depends not on the word's disgust per se but rather on individual sensitivity to disgust. However, the influence of dynamically changing disgust on language comprehension has not yet been researched. In a series of studies, we investigated whether the media's portrayal of COVID-19 will affect subsequent language processing via changes in disgust. The participants were exposed to news headlines either depicting COVID-19 as a threat or downplaying it, and then rated single words for disgust and valence (Experiment 1; N = 83) or made a lexical decision (Experiment 2; N = 86). The headline type affected only word ratings and not lexical decisions, but political ideology and disgust proneness affected both. More liberal participants assigned higher disgust ratings after the headlines *discounted* the threat of COVID-19, whereas more conservative participants did so after the headlines *emphasized* it. We explain the results through the politicization and polarization of the pandemic. Further, political ideology was more predictive of reaction times in Experiment 2 than disgust proneness. High conservatism correlated with longer reaction times for disgusting and negative words, and the opposite was true for low conservatism. The results suggest that disgust proneness and political ideology dynamically interact with perceived environmental safety and have a measurable effect on language processing. Importantly, they also suggest that the media's stance on the pandemic and the political framing of the issue may affect the public response by increasing or decreasing our disgust.

## Introduction

Prior linguistic research has shown that language comprehension is affected by story context, real-world context, and the experience of the listener in the world [1–5]. Furthermore, language processing is affected by the *emotions* it evokes. Affective content of words has been shown to influence how fast they are recognized [6, 7], which also interacts with a range of person-based factors, such as age [8–10], sex [11], character traits and mood [12, 13], native speaker status [14, 15] and others.

Canada (http://www.sshrc-crsh.gc.ca/) Partnership Grant (Words in the World, 895-2016-1008). The funders had no role in study design, data collection and analysis, decision to publish, or preparation of the manuscript.

**Competing interests:** The authors have declared that no competing interests exist.

Two major theoretical approaches to classifying affective words are the dimensional approach and the categorical approach [16]. According to the dimensional approach, words vary along continuous affective dimensions like valence and arousal [17]. Different theoretical models make different predictions about the effect of valence/arousal on lexical processing. The automatic vigilance hypothesis posits rapid allocation of attention to negative stimuli that impedes the processing of other properties of those stimuli, resulting in an inhibitory effect on word recognition and reading [18, 19]. Yet another model proposed by [20, 21] is based on motivated attention and claims that all extremely valenced stimuli regardless of their polarity draw attention faster than neutral stimuli. The empirical evidence has been mixed as well [7, 22–25] and a recent study suggested that valence effects may be modality-specific [26]. The categorical approach usually postulates five discrete universal emotions that words are associated with: happiness, anger, fear, disgust, and sadness [27]. It has been argued that these emotions have at least partially distinct neural correlates [28], and the DISGUST system has been proposed as a primary emotional system eliciting disgust [29]. Importantly, according to [30], discrete emotions influence word processing *over and above* the effects of valence and arousal. In particular, the study found that high-disgust words were processed slower than neutral words when valence and arousal were controlled for.

Given its importance in human evolution and intergroup dynamics, it is not surprising that disgust plays a role in language processing. In fact, several studies showed that the same neural circuitries activate in response to disgusting words as to non-linguistic disgust-related stimuli (facial expressions of disgust, disgusting smells and images, etc.) [31, 32], suggesting that brain areas processing emotional information are not domain-specific [33]. The primary function of disgust is believed to be disease avoidance [34] as healthy individuals have a greater chance to reproduce. For humans, however, disgust extends well beyond these: we can experience aversion to something or someone we deem morally disgusting too, like members of outgroups (e.g., homosexual individuals, immigrants) [35, 36].

### Individual differences in affective word processing

Naturally, people vary in their predisposition to feeling disgusted, which is usually measured by a trait called disgust sensitivity ([37], modified by [38]). Disgust sensitivity is more nuanced than many other character traits because it is less stable and can fluctuate. For instance, events such as epidemics are believed to activate the so-called "behavioral immune system" [39], elevating our disgust sensitivity to protect us from parasites.

There is mounting evidence that listener's character traits, from disgust sensitivity to political ideology, affect various aspects of healthy language processing [e.g., 40–45]. Importantly for our purpose, [46] found that individual disgust sensitivity mediates lexical decision times to high-disgust words. In their study, disgusting words produced an *inhibitory* effect on high disgust-prone participants but a *facilitatory* effect on low disgust-prone participants. This suggests that studying the processing of disgusting words may be meaningless without accounting for individual disgust proneness. The authors explained the effect with the *contextual-learning hypothesis* [47], which claims that the person's response to a specific emotion depends on their life-long experience with this emotion.

Additionally, there is some evidence that traits and fluctuating states may make a differential contribution to affective word processing. For instance, [48] examined the effect of trait anxiety versus an induced anxiety state on negative word recognition and found that individuals in induced anxiety states (but not those scoring high in anxiety trait) showed better recognition of negative words and had a memory bias to negative events. The authors of the study explained it with hypervigilance against danger in a threatening environment that helps an

individual to stay safe. This provides motivation to examine both traits (e.g., disgust sensitivity) and states (e.g., feeling disgusted) in language processing.

## The present study and hypotheses

Prior research has not yet investigated the effect of dynamic changes in a person's disgust on word processing. Since pandemics can naturally elevate our disgust to keep us safe, we aimed to make use of the unique situation the world is currently facing and investigate whether the media's portrayal of the danger of COVID-19 would influence subsequent language processing. Specifically, news platforms often use one of the two approaches in covering the current pandemic: they either emphasize the severity of COVID-19, portraying it as a serious disease demanding public action ("much worse than the flu", "can have lasting detrimental effects on health"), or downplaying it ("no more dangerous than the flu", "kills only the old and the weak"). We investigated whether exposure to these two types of headlines would affect ratings of single words along the dimensions of disgust and valence (Experiment 1) and response times to these words in a lexical decision task (Experiment 2), both as modulated by person-based factors and word-inherent characteristics.

We predicted that being exposed to headlines highlighting the severity of COVID-19 would elevate participants' disgust, helping them to stay safe in a threatening environment. For the rating task, this should translate into higher evaluations on the disgust scale and lower ratings on the valence scale. This effect may potentially be stronger for moderately disgusting words because of the ceiling effect for highly disgusting words. We also expected these effects to be mediated by the participant's baseline disgust sensitivity: people with higher scores should assign higher disgust ratings and be more affected by the headline manipulation than people with lower scores. Combining a word rating task with a lexical decision task allowed us to examine both the fluctuations of participant's disgust as reflected in their word ratings and how these fluctuations affect the core language processing mechanisms, such as lexical access. Based on the findings by [46], we expected that 1) less disgust-prone participants would have shorter lexical decision times for disgusting words, whereas high disgust-prone participants would have longer lexical decision times for the same words 2) the severe headlines would exacerbate this effect due to increased disease salience.

We additionally examined whether the effect of headlines is mediated by the participant's political ideology. Conservatives tend to score higher on disgust in general [49–51], and making people physically disgusted shifts their attitudes to the conservative end of the political spectrum [52]. One would thus expect conservative views to strengthen as a function of physical disgust and/or perceived disease vulnerability. Indeed, a study by [53] conducted in the U. S. and Poland found that exposure to press reports on COVID-19 increased the support for conservative presidential candidates (see [54] for similar findings during the Ebola outbreak). At the same time, high politicization of COVID-19 in the U.S. has led to more denial of the virus and less adherence to social distancing guidelines by the conservative compared to the liberal [55]. Largely, this attitude stems from the framing of the issue by conservative politicians as a trade-off between economic growth and saving human lives, with a common rhetoric that the cure "cannot be worse than the problem" [56]. This has created a paradox whereby conservatives downplay the threat of COVID-19 more and engage in social distancing less than liberals. It is thus difficult to predict how the headline manipulation would interact with political ideology in our study. We had two hypotheses regarding the conservative response to the severe headlines: 1) Higher conservatism scores will correlate with higher disgust ratings due to higher disgust proneness, and highlighting the threat of COVID-19 will exacerbate this effect; 2) Headlines emphasizing the threat will be considered uncredible, untrustworthy, and

be discarded by more conservative participants, resulting in no effect in disgust ratings. For liberals, we predicted that the severe headlines will not produce a strong effect due to them accepting this situation as a norm, whereas headlines downplaying COVID-19 will produce a strong disgust reaction due to a heightened sense of threat and a conflict with their own view of the severity of the virus. This should result in higher disgust ratings following the downplaying headlines.

We should also note that while most of the data about politicization and polarization of the COVID-19 pandemic does come from the U.S., [57, 58] did not find significant differences between the U.S. and Canada in this respect (whereas the level of political polarization was significantly lower in the U.K.). We thus assumed that the level of politicization and polarization of the COVID-19 pandemic in Canada is high.

## Experiment 1—Word ratings

### Materials and methods

**Participants.**   Eighty-three students at the University of Alberta received partial course credit for their participation. Nineteen were removed from the data analysis (14 = indicated they wished their data to be withdrawn, 5 = provided incorrect answers to trap questions in the Disgust Sensitivity survey), leaving 64 participants in the data analysis (43F [67%], mean age = 20.3, range = 17–41, SD = 3.2). Thirty-four were native speakers of English. All participants who did not choose English as the first language they acquired in childhood were classified as non-native speakers. The participants were asked to rate their English proficiency on a 5-point scale. Mean self-reported proficiency for non-native speakers was 4.0/5 (SD = 0.84) and for native speakers 4.9/5 (SD = 0.40). The plan for this study was reviewed for its adherence to ethical guidelines by a Research Ethics Board at the University of Alberta (reference number Pro00102348).

**Materials.**   We used two databases: norms of valence, arousal, and dominance (VAD) by [17] and the NRC Emotion Intensity Lexicon by [59]. The latter provides emotions with which the words are associated and the strength of association scores. Only words that had disgust scores associated with them were extracted (1,093 in total). From these, we selected the words that also occurred in the former database. This left us with 787 words. Since the VAD and disgust scores were on different scales, we rescaled them uniformly. We kept arousal values constant (between -1 and 1 in a normalized distribution) and split the words into four subsets (high/low valence high/low disgust). Because there were virtually no high valence/high disgust items, we excluded this group. This allowed us to keep stricter thresholds for the other three categories and thus get better representatives of those categories:

- high disgust ($>$ .75) low valence ($<$ -.75);

- low disgust ($<$ -.75) high valence ($>$ .75);

- low disgust ($<$ -.5) low valence ($<$ -.5).

For each category, we selected 33 words, 99 words in total (see Table 1). Correlations between words' characteristics together with their *p*-values are provided in Table 2. We took concreteness ratings from [60], who used [61] as training data. Age of acquisition was taken from [62].

For the headline manipulation, we randomly selected seven news articles emphasizing the severity of COVID-19 and eight articles downplaying it. We then made screenshots of the headlines, some of which also had illustrations. No disturbing imagery was used. In the severe condition, the images showed patients in hospitals surrounded by healthcare workers in

**Table 1. Mean characteristics of words by category.**

| Word category | Valence | Arousal | Disgust | Log. Freq | Length | Concreteness | AoA |
|---|---|---|---|---|---|---|---|
| high disgust low valence | -1.09 (.29) | .15 (.57) | 1.39 (.42) | 7.93 (1.43) | 7.18 (2.31) | 4.11 (2.19) | 9.61 (2.36) |
| low disgust high valence | 2.05 (.88) | -.11 (.54) | -1.61 (.50) | 8.92 (1.70) | 6.06 (2.09) | 3.85 (2.67) | 7.98 (2.58) |
| low disgust low valence | -.74 (.20) | -.01 (.45) | -.90 (.36) | 8.25 (1.22) | 8.06 (2.60) | 1.23 (1.04) | 8.31 (1.71) |
| total | .07 (1.51) | .01 (.53) | -.37 (1.36) | 8.36 (1.51) | 7.10 (2.46) | 3.06 (2.44) | 8.63 (2.33) |

AoA = Age of Acquisition. SDs are in brackets.

protective gear, an illustration of a SARS-CoV-2 virion, or an infographic. The downplayed condition included images of people protesting the lockdown or enjoying their vacation despite the pandemic. The stimuli are available in Appendix 1 in S1 File and the headlines in Appendix 2 in S1 File.

**Procedure.** The experiment was programmed in PsychoPy3 [63] and conducted online on the Pavlovia platform at pavlovia.org. Each session started with newspaper headlines appearing on the screen one by one. Depending on the condition, the headlines either highlighted the severity of COVID-19 or downplayed it. The headlines were only shown in the beginning of each session, not before every trial. After the participant looked through all the assigned headlines, the main experiment began. Single words appeared on the screen one after another, and the participant's task was to rate how disgusting each word feels to them (1 = "not at all" to 5 = "extremely") and how positive/negative it feels (1 = "very negative" to 5 = "very positive"). Three questionnaires were presented after the main experiment: The Disgust Sensitivity Revised scale (DS-R) ([37], modified by [38]), the Wilson-Patterson (W-P) Conservatism Scale [64], and a short language background questionnaire. Before submitting their data, each participant was explicitly asked whether they wanted to withdraw their data.

**Data analysis.** All the stimuli, raw data, and scripts used in this experiment are available on Open Science Framework. Initially the data was analyzed using generalized additive mixed modeling (GAMM) for ordinal data following [65]. However, the resulting analysis was highly complex, and it was advised that we use linear mixed modeling instead. The results of the two analyses were virtually identical, and we will thus report linear mixed modeling for simplicity. The full GAMM analysis with the scripts and plots is available for review on OSF. Linear mixed-effects regression models were fitted using the *lmer* function from the *lme4* package (1.1–27.1) [66]. *P*-values were obtained using the *sjPlot*::*tab_model* function from the *sjPlot* package (2.8.9) [67]. The results were plotted with the same package. The scores in the DS-R and W-P questionnaires were standardized (by subtracting the mean and dividing the

**Table 2. Correlation coefficients between stimuli's characteristics.**

| | Valence | Arousal | Disgust | Log. Freq | Length | Concreteness | AoA |
|---|---|---|---|---|---|---|---|
| **Valence** | 1.00 | -.21* | -.73*** | .29** | -.28** | .15 | -.28** |
| **Arousal** | -.21* | 1.00 | .21* | .21* | .23* | -.08 | .01 |
| **Disgust** | -.73*** | .21* | 1.00 | -.31** | .08 | .21* | .40*** |
| **Log. Freq** | .29** | .21* | -.31** | 1.00 | -.04 | -.12 | -.36*** |
| **Length** | -.28** | .23* | .08 | -.04 | 1.00 | -.44*** | .33*** |
| **Concreteness** | .15 | -.09 | .21* | -.12 | -.44*** | 1.00 | -.09 |
| **AoA** | -.28** | .01 | .40*** | -.36*** | .33*** | -.09 | 1.00 |

AoA = Age of Acquisition. Asterisks indicate significance levels (*** < .001, ** < .01, * < .05).

remainder by standard deviation); word frequency was log-transformed. The models included a maximal supported random structure (random intercepts for subjects and items; by-subject slopes were tested but the models did not converge). Since the correlation between a word's inherent disgust and valence was high (r = -.73), we additionally tested each fitted model using the *check_collinearity()* function from the *performance* package (0.8.0) [68]. The function provides Variance Inflation Factors (VIF) for each term in a model. All VIFs were < 2.5 for both disgust and valence rating models, so none of them had collinearity issues. The correlation between the DS-R and W-P scores was .17. Cronbach's alpha was .74 for the DS-R inventory and .80 for the W-P inventory, indicating good questionnaire reliability.

**Results.**  Since disgust sensitivity is not a stable measure, presenting the DS-R survey before or after the experiment had its pros and cons. Completed before the experiment, the questionnaire may prime the participants beyond the effect of headlines, or they may guess the purpose of the study. There is also evidence that mere exposure to disgust surveys before the experiment increases temporal disease salience and intergroup bias [69]. If the survey is completed after the experiment, the scores may be shifted, especially if the participants were exposed to the severe headlines. Since we opted for post-experiment surveys, we checked the mean and the range of DS-R scores per condition to detect any abnormalities. The mean for the downplayed condition was 59.5 (range 22 to 80), and for the severe condition 62.8 (range 26 to 88). A Student's t-test showed that the means did not differ significantly (*t* = -0.84, df = 62, *p*-value = .404) and thus were unlikely to present a confound.

The correlation between disgust and valence ratings was -.48. This was expected since word disgust and valence were strongly correlated (-.73).

*Disgust rating.* We fitted a fully specified model with the following predictors: control variables (log frequency, word arousal, word valence, native speaker (ns) status), words' characteristics (word disgust), headline condition, participants' individual differences (DS-R, W-P) and two- and three-way interactions between all the predictors excluding control variables. The results are provided in Table 3.

All control variables were insignificant. Native speaker status did not have any effect on word ratings. There were expected main effects of word disgust and participants' disgust sensitivity. More disgust-prone participants rated everything as more disgusting, and more disgusting words were rated as more disgusting. These effects were meaningful even in the presence of higher-level interactions with these variables. We will now go over all the significant interactions in detail.

Political ideology interacted significantly with the headline (Fig 1A) and with word disgust (Fig 1B) (the three-way interaction was not significant). Fig 1A supports our prediction about the politicization and polarization of the COVID-19 pandemic. The plot shows that the headlines produced the exact opposite effect on participants depending on what side of the conservatism scale they were on. More liberal participants (lower W-P score) rated the stimuli as more disgusting in the downplayed compared to the severe condition. In contrast, the effect of the headline was reversed and stronger for more conservative participants. The severe headlines made them rate the stimuli as substantially more disgusting compared to the downplayed condition. Fig 1B shows that political ideology also affected baseline disgust ratings. On average, less conservative participants assigned more extreme disgust ratings compared to more conservative participants, rating high-disgust words substantially higher and low-disgust words slightly lower. We are not aware of any previous studies that investigated the effects of political ideology on word ratings. This is a novel finding that warrants further research.

Two two-way interactions with word disgust were significant: DS-R x word disgust and headline x word disgust. In addition, the three-way interaction between headline, word disgust, and DS-R was significant (Fig 2). Fig 2A shows that DS-R and word disgust interacted in

**Table 3. Summary of the linear mixed-effects model with disgust rating as a dependent variable.**

| Predictors | Estimates | CI | p | |
|---|---|---|---|---|
| (Intercept) | 2.98 | 2.59–3.38 | < .001 | *** |
| log freq | 0.01 | -0.04–0.05 | .794 | |
| ns [ns] | -0.07 | -0.31–0.18 | .598 | |
| wd arousal | 0.02 | -0.09–0.13 | .732 | |
| wd valence | -0.02 | -0.07–0.04 | .511 | |
| wd disgust | 0.56 | 0.50–0.62 | < .001 | *** |
| headline [covid severe] | 0.11 | -0.12–0.34 | .332 | |
| DS-R | 0.27 | 0.11–0.43 | .001 | ** |
| W-P | -0.11 | -0.27–0.05 | .166 | |
| headline [covid severe] * wd disgust | -0.05 | -0.08--0.01 | .012 | * |
| headline [covid severe] * W-P | 0.25 | 0.01–0.48 | .038 | * |
| headline [covid severe] * DS-R | -0.06 | -0.30–0.18 | .619 | |
| DS-R * wd disgust | 0.08 | 0.06–0.11 | < .001 | *** |
| W-P * wd disgust | -0.07 | -0.10--0.05 | < .001 | *** |
| DS-R * headline [covid severe] * wd disgust | -0.07 | -0.10--0.03 | < .001 | *** |
| W-P * headline [covid severe] * wd disgust | -0.01 | -0.05–0.03 | .648 | |
| **Random Effects** | | | | |
| σ2 | 0.99 | | | |
| τ00 word | 0.06 | | | |
| τ00 participant | 0.20 | | | |
| ICC | 0.21 | | | |
| N participant | 64 | | | |
| N word | 99 | | | |
| Observations | 6336 | | | |
| Marginal R2 / Conditional R2 | 0.339 / 0.478 | | | |

Model's formula: disgust rating ~ log freq + wd arousal + wd valence + ns + DS-R*headline*wd disgust + W-P*headline*wd disgust + (1 | participant) + (1 | word).
Asterisks indicate significance (*** < .001, ** < .01, * < .05, . < .07).

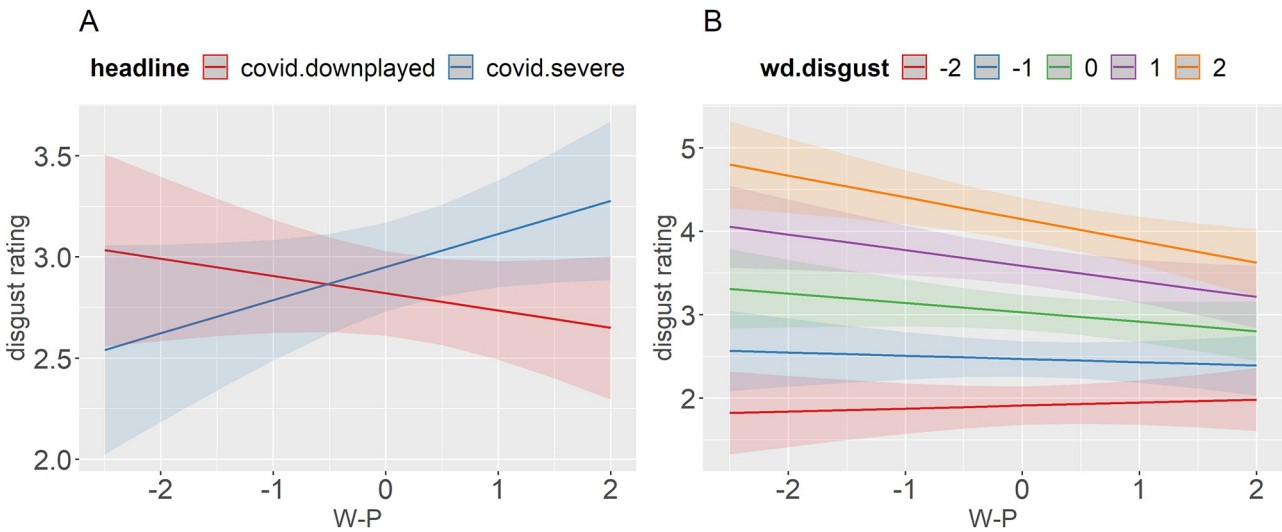

**Fig 1. Predicted values (marginal effects) of the interaction between W-P and headline (A) and W-P and word disgust (B) with disgust rating as a dependent variable.**

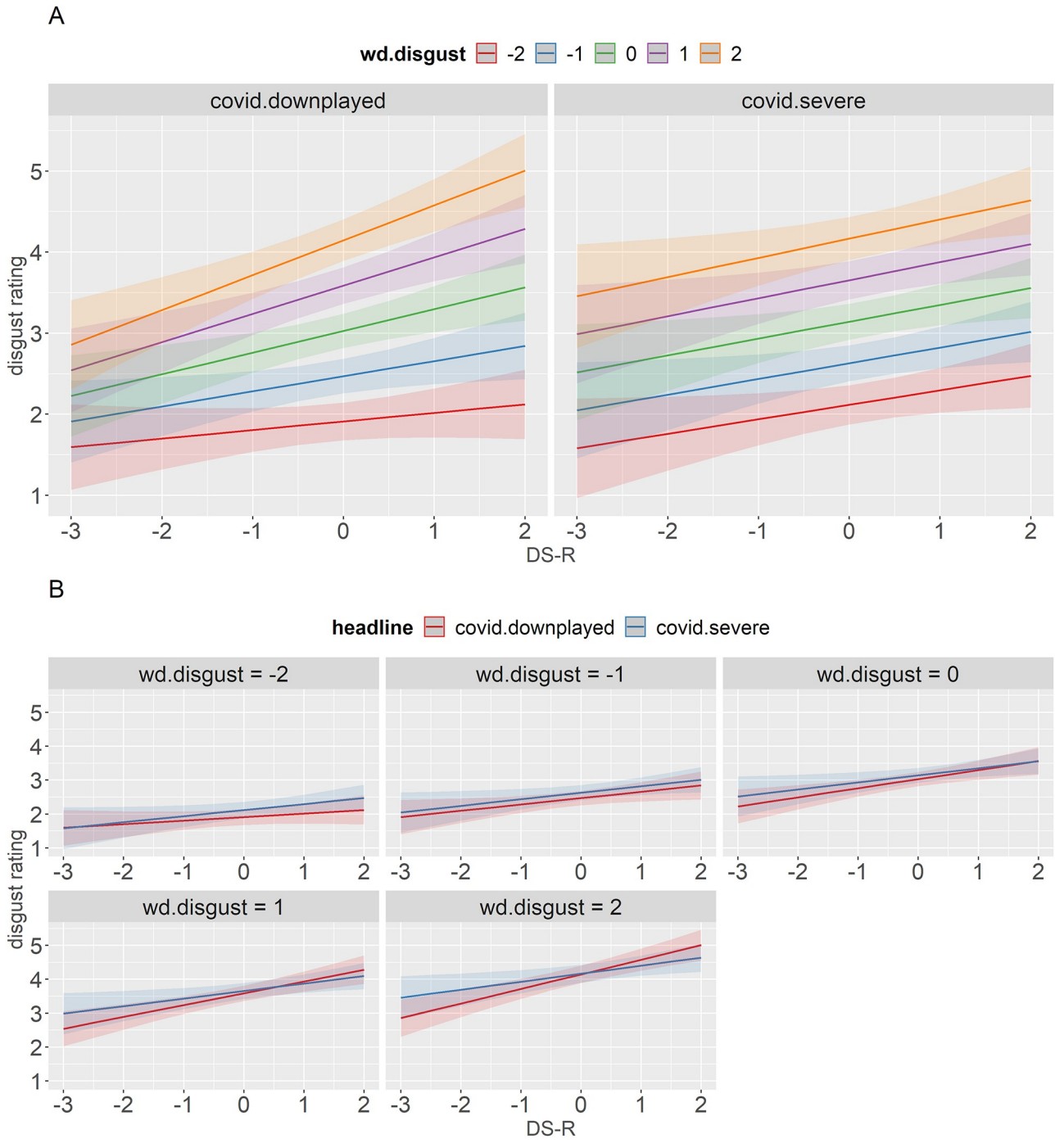

**Fig 2. Predicted values (marginal effects) of the interaction between DS-R, word disgust, and headline with disgust rating as a dependent variable.** Panel A shows a breakdown by headline, Panel B shows a breakdown by word disgust.

the downplayed condition only. The lines for the quantiles of word disgust are virtually parallel following the severe headlines, reflecting main effects of word disgust and DS-R, whereas they are fan-shaped following the downplaying ones. High-disgust words (disgust index > 0) had a much steeper slope than low-disgust words (disgust index < 0) in the downplayed condition.

This suggests that downplaying the threat of the virus has a larger effect for more disgusting stimuli if the participant is more sensitive to disgust. Moving a little higher on the DS-R scale in this case leads to a substantial increase in disgust ratings for high-disgust words but a much smaller increase for low-disgust words. Fig 2B additionally shows that low disgust-prone participants consistently rated words higher for disgust in the severe condition compared to the downplayed one, and the more disgusting the word, the larger was an increase compared to the rating in the downplayed condition. This means that the headline manipulation was successful. The results for high disgust-prone participants are more complicated. In the severe condition, they rated low-disgust words higher for disgust but high-disgust words *lower* for disgust.

*Valence rating*. We fitted the same model as for disgust rating, with the only difference that word disgust was now a control variable and word valence was examined in interaction with all the other predictors. The results are provided in Table 4.

Naturally, word valence was directly correlated with participants' ratings: the more positive the word, the more positively it was rated, and vice versa. This effect also remained in the presence of higher-level interactions. Word arousal was marginally significant. On average, words with lower arousal were rated as slightly more positive, and vice versa. This is in line with a

**Table 4. Summary of the linear mixed-effects model with valence rating as a dependent variable.**

| Predictors | Estimates | CI | p | |
|---|---|---|---|---|
| (Intercept) | 2.26 | 1.93–2.59 | < .001 | *** |
| log freq | -0.00 | -0.04–0.04 | .897 | |
| ns [ns] | 0.03 | -0.08–0.15 | .591 | |
| wd arousal | -0.11 | -0.21–0.00 | .051 | . |
| wd disgust | -0.03 | -0.09–0.02 | .271 | |
| wd valence | 0.49 | 0.44–0.54 | < .001 | *** |
| headline [covid severe] | 0.01 | -0.10–0.11 | .927 | |
| DS-R | -0.15 | -0.23–-0.08 | < .001 | *** |
| W-P | 0.10 | 0.02–0.17 | .014 | * |
| headline [covid severe] * wd valence | -0.02 | -0.05–0.00 | .053 | . |
| headline [covid severe] * W-P | -0.00 | -0.11–0.11 | .950 | |
| headline [covid severe] * DS-R | 0.09 | -0.03–0.20 | .134 | |
| DS-R * wd valence | 0.05 | 0.03–0.07 | < .001 | *** |
| W-P * wd valence | -0.06 | -0.07–-0.04 | < .001 | *** |
| DS-R * headline [covid severe] * wd valence | -0.03 | -0.05–-0.00 | .032 | * |
| W-P * headline [covid severe] * wd valence | 0.01 | -0.01–0.04 | .329 | |
| **Random Effects** | | | | |
| σ2 | 0.53 | | | |
| τ00 word | 0.06 | | | |
| τ00 participant | 0.04 | | | |
| ICC | 0.16 | | | |
| N participant | 64 | | | |
| N word | 99 | | | |
| Observations | 6336 | | | |
| Marginal R2 / Conditional R2 | 0.491 / 0.573 | | | |

Model's formula: valence rating ~ log freq + wd arousal + wd disgust + ns + DS-R*headline*wd valence + W-P*headline*wd valence + (1 | participant) + (1 | word).
Asterisks indicate significance (*** < .001, ** < .01, * < .05, . < .07).

weak negative correlation between word valence and word arousal in our experiment (r = -.21, see Table 2). We also found main effects of both disgust sensitivity and political ideology, which were meaningless due to higher-level interactions.

A two-way interaction between DS-R and word valence as well as a three-way interaction between headline, DS-R, and word valence were significant. Fig 3A corroborates the effect found for disgust rating. Once again, word valence mostly interacted with DS-R in the down-played condition, with the lines having a distinct fan shape. Words with the lowest valence had a much steeper slope than words with the highest valence, suggesting that individual disgust sensitivity has a much stronger effect on low- than on high-valence words. Fig 3B shows that, same as for disgust, the least disgust-prone individuals rated words as more negative in the severe compared to the downplayed condition, and this effect was the largest for the most neg-ative words. Most disgust-prone participants, however, again rated negative stimuli as more negative in the *downplayed* compared to the severe condition. It should be noted that the dif-ference between the two ends of the DS-R scale was very small, even for the most negative words.

The interaction between the participant's political ideology and word valence (Fig 4) cor-roborated and further extended the effect found for disgust ratings (Fig 1B). On average, more liberal participants rated all positive words as more positive and negative words as more nega-tive regardless of the condition. The joint findings from the disgust and valence ratings essen-tially translate to less conservative participants being more extreme with their ratings and having a broader range of responses.

**Discussion.** As predicted, the headlines affected the participant's disgust ratings differ-ently depending on their political ideology and disgust proneness. Less conservative partici-pants rated the stimuli higher for disgust following the downplaying headlines, whereas more conservative participants assigned higher disgust ratings following the severe headlines. We can think of two possible explanations for these results, both of which reflect the dominant COVID-19 narrative in the two political spheres [55, 58, 70]. The first, and the simplest, expla-nation is that the stance representing one's political outgroup (headlines emphasizing the threat of the virus for the conservative participants and headlines downplaying it for the liberal participants) evokes more disgust and negative affect. Since hardly any language processing is done without affective evaluation [71], it is possible that a take on the virus that is so far from your own would elicit a strong emotional response. However, in this case, one would also logi-cally expect not only higher disgust but also more negative ratings. This was not the case, and the interaction between political ideology and headline was not significant. The second and more nuanced explanation would be habituation of disgust and thus an asymmetric response to the severe headlines. Since the dominant and largely supported view by the liberal politi-cians in Canada revolves around the high danger of the virus, this stance may result in desensi-tization to it by the liberal public and a consequent lack of strong disgust response. The opposite is true for more conservative participants, with the dominant conservative view being the lack of such danger. Importantly, the severe headlines did elevate disgust levels in more conservative participants despite the lack of trust in contradictory media [55] and science in general [70] found for conservative participants in previous research. Again, two explanations are possible: 1) more conservative participants did not treat the severe headlines as unreliable, or 2) more conservative participants did consciously register the severe headlines as unreliable but their bodies still reacted to the increased disease salience by elevating their disgust levels. It is unfortunately not possible at the time to choose between the two explanations.

Further, we found initial evidence that subscribing to a more liberal or more conservative ideology may be correlated with the extremity of ratings. Political ideology affected word rat-ings for both disgust and valence regardless of the condition, with less conservative

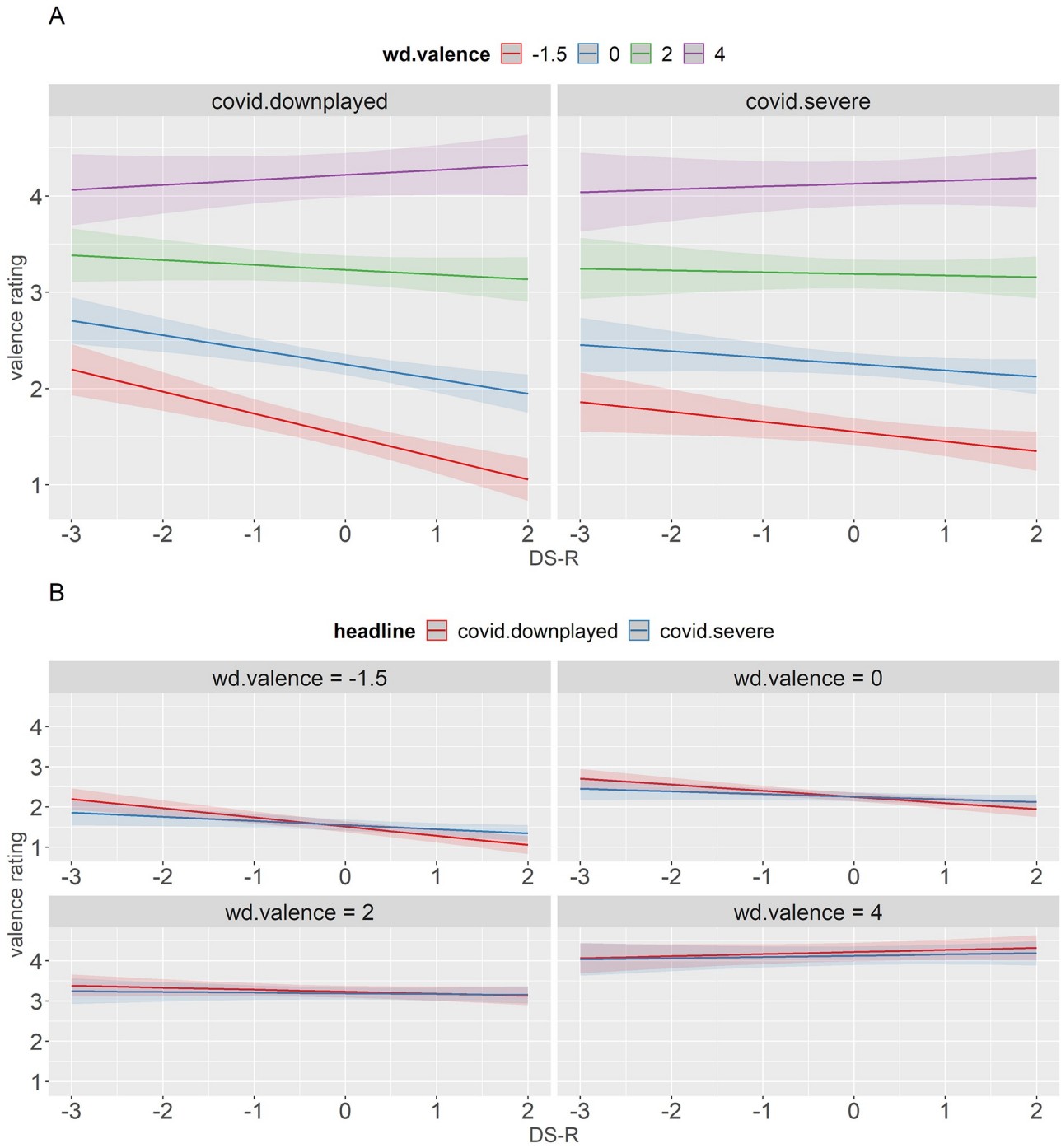

**Fig 3. Predicted values (marginal effects) of the interaction between DS-R, word valence, and headline with valence rating as a dependent variable.** Panel A shows a breakdown by headline, Panel B shows a breakdown by word valence.

participants providing more extreme ratings (higher disgust ratings for disgusting words and lower for non-disgusting words as well as higher valence ratings for positive words and lower for negative words). To the best of our knowledge, this is a novel finding. Individual differences in average ratings (the so-called "rater's generosity") was previously found to affect

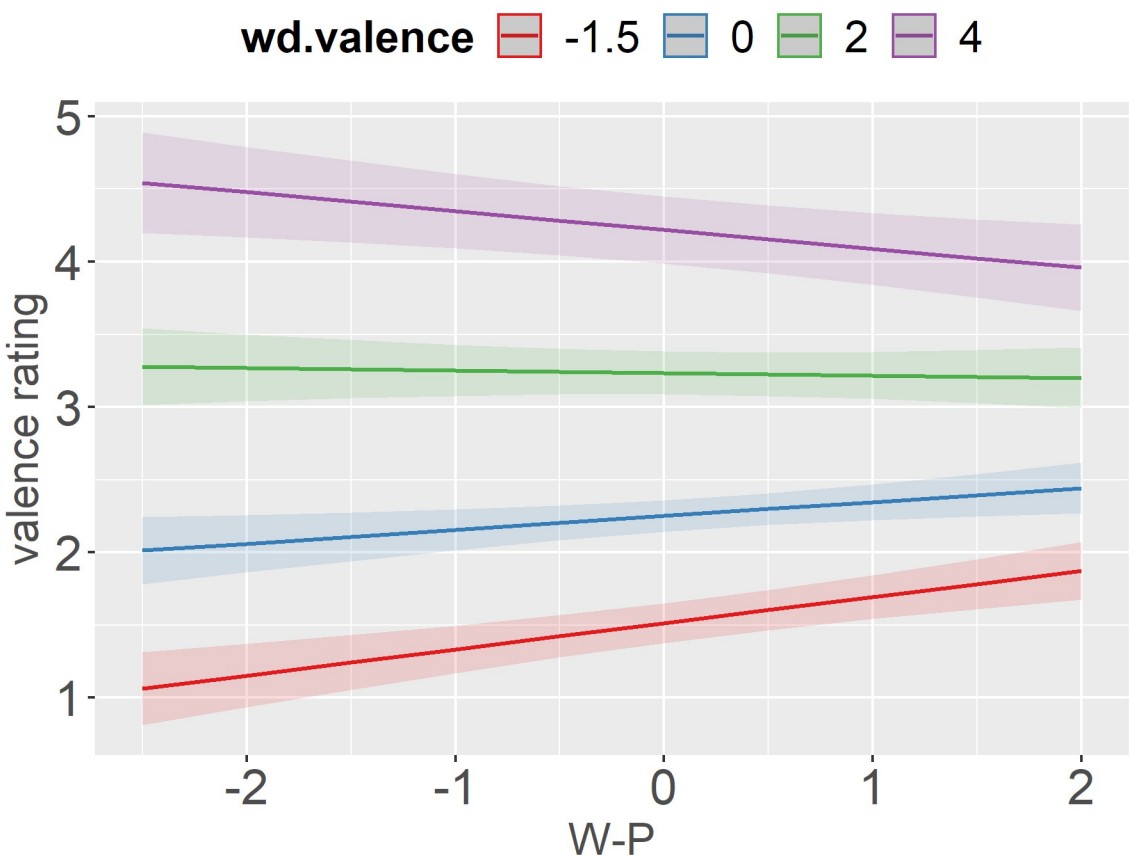

**Fig 4. Predicted values (marginal effects) of the interaction between W-P and word valence with valence rating as a dependent variable.**

acceptability ratings both by itself and in interaction with other predictors [65], but we are not aware of any research on factors affecting ratings' range/extremity. At the very least, our findings highlight the importance of accounting for individual differences in rating studies, in particular for political ideology that has been by and large ignored in linguistic research.

Individual disgust sensitivity affected word ratings both by itself and in interaction with other predictors. Overall, disgust proneness was correlated with higher disgust ratings. This was expected from the definition of disgust proneness and, importantly, shows that word ratings for disgust can reliably serve as a proxy to participant's disgust. Disgust sensitivity also interacted with word disgust and the headlines. Thus, downplaying the threat of the virus had the largest effect for the most disgusting and negative words: a slight increase in participant's disgust proneness was associated with a significant increase in word ratings for disgust and negativity. In addition, whereas low disgust-prone participants consistently rated all the words as more disgusting in the severe condition, only low and moderately disgusting words, but not highly disgusting ones, were rated higher for disgust in the same condition by high disgust-prone participants. The same effect was observed for valence ratings. High disgust and low valence words were actually rated lower for disgust and higher for valence in the severe compared to the downplayed condition by high disgust-prone participants. While the reasons for this effect are not entirely clear, it should be noted that the effect itself was very small and that high disgust-prone participants rated high disgust and low valence words very close to the top

of the disgust scale and the bottom of the valence scale in both conditions. This may potentially suggest a ceiling effect.

All in all, the results of this study show that individual differences such as disgust proneness and political ideology dynamically interact with how safe the environment around the participant feels and have a measurable effect on language processing, specifically on word ratings. In the next experiment, we intended to find out whether this effect also extends to online language processing.

## Experiment 2—Lexical decision

### Materials and methods

**Participants.** Eighty-six students at the University of Alberta received partial course credit for their participation. None participated in Experiment 1. Seventeen participants were removed from the data analysis (7 = indicated they wished their data to be withdrawn, 8 = provided incorrect answers to trap questions in the Disgust Sensitivity survey, 2 = mean reaction times were above 3500 ms). This left us with sixty-nine participants (51F [83%], mean age = 20.7, range 17–59, SD = 5.6). Forty-two were native speakers of English. Participants who did not choose English as the first language they acquired in childhood were classified as non-native speakers. The participants were asked to rate their English proficiency on a 5-point scale. Mean self-reported English proficiency for non-native speakers was 4.0/5 (SD = .83) and for native speakers 5.0/5 (SD = .15). The plan for this study was reviewed for its adherence to ethical guidelines by a Research Ethics Board at the University of Alberta (reference number Pro00102348).

**Materials.** The real English words were the same as in Experiment 1. Ninety-three pseudowords were created by modifying one or more letters in existing English words. A native speaker of English checked the final list of pseudowords and made sure they did not contain any real but archaic words, words that look like a typo in real words, slang words, and words that do not currently exist but sound like they could be a neologism. The full list of the stimuli, including the pseudowords, is available in Appendix 1 in S1 File.

**Procedure.** A visual lexical decision task was used. Each trial began with a letter string appearing in the center of the computer screen and staying until the participant responded. The participants indicated whether the letter strings were real English words or not by pressing either the left or the right arrow (the side of "word"/"non-word" arrows was counterbalanced between participants). The instruction was to respond as quickly and as accurately as possible. Seven practice trials preceded the experimental trials, with feedback after each. No feedback was given in the main session. Opportunities for taking a break were given after each third of the stimuli. Otherwise, the procedure was the same as in Experiment 1.

**Data analysis.** Overall mean accuracy for words was 91.2% and for non-words 85.5%. For native speakers, mean accuracy for words and non-words was 96.3% and 89.8%, respectively. For non-native speakers, mean accuracy for words and non-words was 83.7% and 79.2%, respectively.

Before data analysis, pseudowords, incorrect responses (8.8%), two participants with mean RTs over 3500 ms (2.4%), and responses below 100 ms and above 2500 ms (1.9%) were removed. In addition, we removed trials with RTs below and above 2.5 SDs per participant (3%). The reaction times were reciprocally transformed (-1000/RTs), the right tail was removed (0.4%), and the RTs were multiplied by 1000 to avoid extremely small numbers [72]. This means that smaller numbers (i.e., bigger negative numbers) in the final analysis correspond to shorter lexical decision times, so the plots can be read intuitively. One more predictor, match/mismatch of the participant's dominant hand and the location of the "word"

button, was added to the model. The model included a maximal supported random structure (random intercepts for subjects and items). The by-subject slope for trial was tested and produced a singular fit, suggesting an overfitted model, so we removed it. Compared to Experiment 1, a few more control variables were used: age of onset, concreteness, and orthographic length (all standardized). The correlation between the participant's W-P and DS-R scores was higher (r = .4) than in Experiment 1. Same as before, we tested the fitted fully specified model using the *check_collinearity()* function from the *performance* package (0.8.0). The results showed several potential collinearity issues (VIFs for word disgust, word valence, DS-R, and W-P were all in the 3.5–4 range; VIFs for interactions were in the 5–8 range even though the predictors were standardized). Testing word disgust and word valence in separate models did not remove multicollinearity to a sufficient extent (VIFs for DS-R and W-P were still > 3.7). We thus had to run four different models: for word disgust and DS-R, for word disgust and W-P, for word valence and DS-R, for word valence and W-P. This resolved multicollinearity (all VIFs < 2.7). Cronbach's alpha was .78 for the DS-R inventory and .83 for the W-P inventory, again indicating good reliability.

**Results.** We first checked the range and means of disgust sensitivity scores depending on the headline condition. The mean for the downplayed condition was 63.4 (range 25 to 91), and for the severe condition 62.1 (range 21 to 89). Student's t-test confirmed that the means were not significantly different ($t = 0.52$, df = 67, p-value = .605).

The outputs of the four models are available in Appendices 3–6 in S1 File. In all the models, there were significant main effects of log frequency, age of acquisition, and native speaker status. Consistent with previous research, higher frequency words were associated with shorter RTs than lower frequency words. Also in line with previous research, non-native speakers had longer reaction times than native speakers [14, 15]. Age of acquisition affected RTs in a predictable direction: words acquired earlier were reacted to faster. In addition, concreteness was only significant in the models with word disgust: less concrete words were recognized slightly faster than more concrete ones.

We will now go over all the interactions. First, there were significant two-way interactions W-P x word disgust and W-P x word valence. Fig 5A shows that the effect of word disgust changed direction depending on the side of the conservatism scale. More liberal participants recognized high-disgust words faster than low-disgust words. In contrast, the more disgusting the word, the longer it took more conservative participants to recognize it. This is in line with the findings for disgust proneness by [46], since conservatism and disgust proneness have been found to be correlated both in prior research [49–51] and this experiment (r = .4). Fig 5B corroborates this effect for word valence. While more liberal participants had shorter RTs to negative words, more conservative participants had longer RTs to negative words. The opposite was true for positive words.

In addition, there was a significant two-way interaction DS-R x word disgust and a marginally significant interaction DS-R x word valence (p = .06). As can be seen in Fig 6A, we corroborated the direction of the findings of [46], although the effect was very small for high disgust-prone participants. The least disgust-prone participants reacted faster to disgusting than to non-disgusting words, and this facilitating advantage decreased as the participant moved a little higher on the DS-R scale. The marginally significant interaction with word valence was in the same direction (Fig 6B): negative words produced a facilitatory effect on low disgust-prone participants compared to positive words.

Since political ideology and disgust sensitivity affected RTs consistently (disgusting and negative words had a facilitatory effect on more liberal and on less disgust-prone participants), it is important to know which predictor, W-P and DS-R, explained more variance. When W-P and DS-R were added together in the models for word disgust and word valence, only W-P

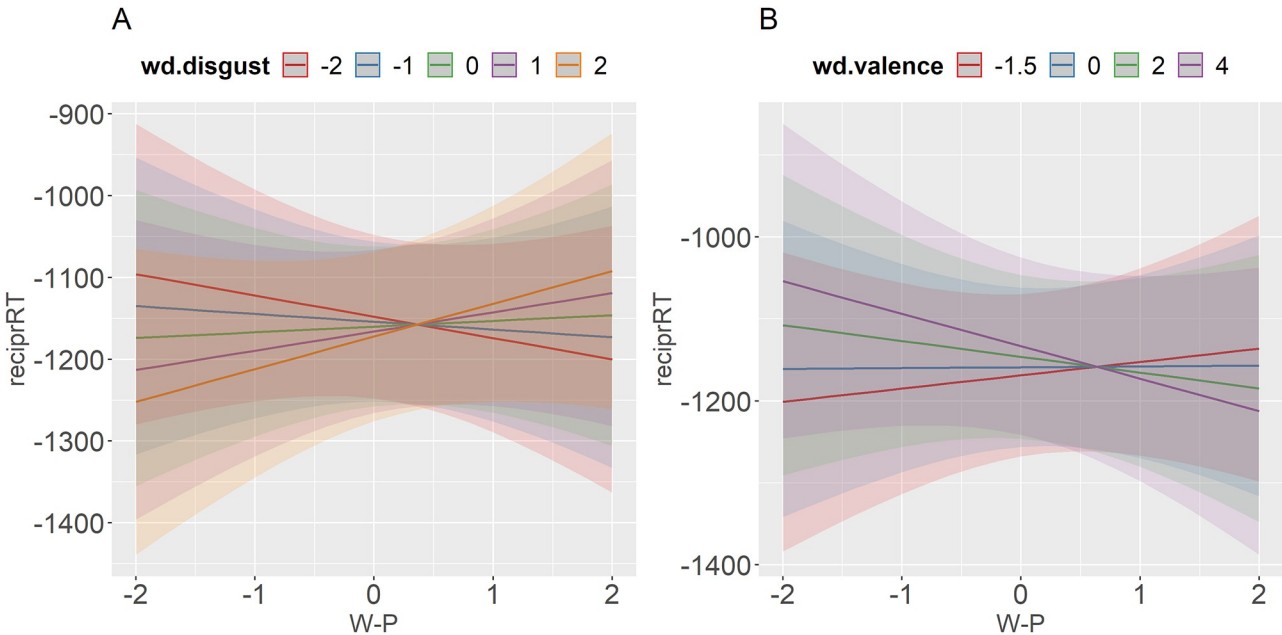

**Fig 5. Predicted values (marginal effects) of the interaction between W-P and word disgust (A) and W-P and word valence (B) with reciprocally transformed RTs as a dependent variable.** Bigger negative numbers correspond to shorter lexical decision times.

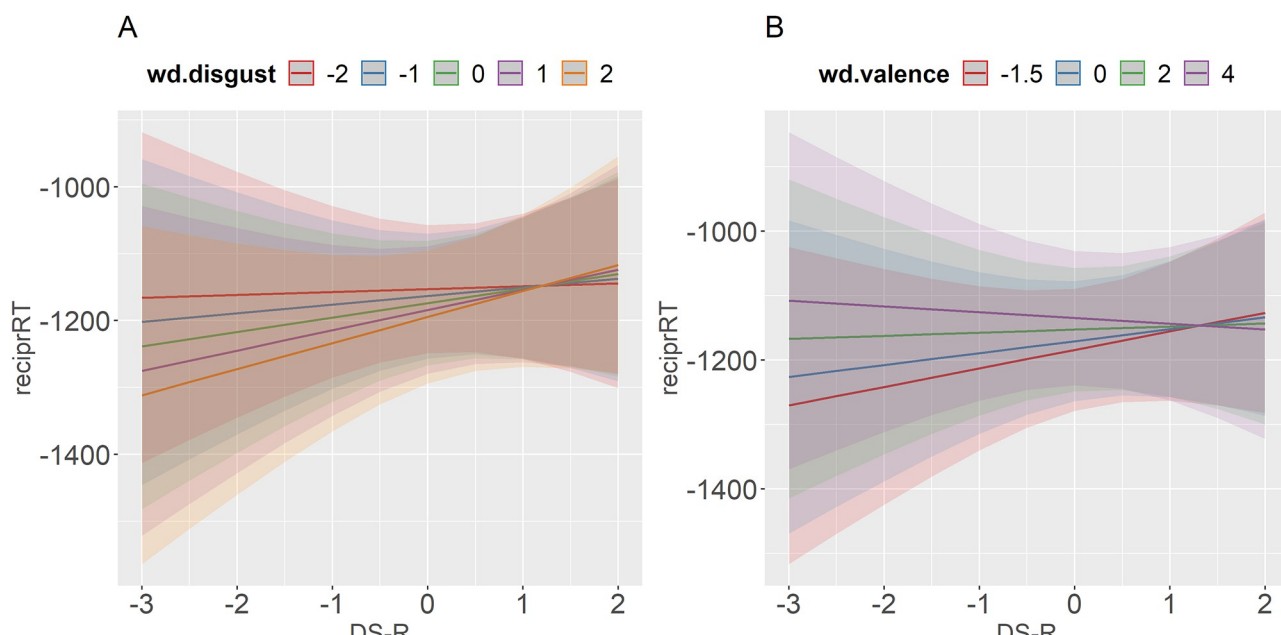

**Fig 6. Predicted values (marginal effects) of the interaction between DS-R and word disgust (A) and DS-R and word valence (B) with reciprocally transformed RTs as a dependent variable.** Bigger negative numbers correspond to shorter lexical decision times.

came out significant (word disgust x W-P: $p < .001$; word valence x W-P: $p = .014$). Stepwise backward elimination using the *step()* function from the *lmerTest* package (3.1–3) [73] showed that, apart from age of onset, native speaker status, and log frequency only the word disgust x W-P interaction significantly improved the model's fit. Thus, political ideology was more predictive of RTs in this study than disgust sensitivity.

Importantly, no effect of headline was found, either by itself or in interaction with other predictors.

**Discussion.**   The main finding of this experiment was the effect of political ideology on word recognition latencies. Lower conservatism was associated with *faster* recognition of disgusting and negative words, whereas higher conservatism was associated with *slower* recognition of disgusting and negative words, regardless of the headline. Assuming that disgust and conservatism are correlated, this is in line with the findings of [46]. Importantly, however, the effect of disgust proneness disappeared when the two predictors were added to the models together. This suggests that political ideology was more important in predicting RTs than disgust proneness. This has important implications for future studies using the lexical decision task, especially with words varying in disgust and valence. Moreover, removing non-native speakers did not change the results for political ideology (interactions of W-P with both word disgust and valence were significant and in the same direction) but removed the effect of disgust proneness. This once again testifies to the stability of the results for political ideology and provides evidence that combining native and non-native speakers did not have any substantial effect on the results.

Importantly, we did not find any effect of the headlines on word recognition latencies. This finding, although unexpected, is not in direct contradiction to the results of Experiment 1, where such an effect was observed. The lexical decision task and the rating task reflect very different dimensions of processing. The lexical decision task indexes the ease of accessing a word in long-term memory, whereas ratings are an explicit affective evaluation that is not time-sensitive. Thus, the results of this experiment suggest that fluctuating levels of disgust may not modulate the accessibility of disgust-related concepts in long-term memory while still modulating their affective evaluation.

## General discussion

Our study tested the hypothesis that the media's stance on the pandemic may elevate or reduce participants' disgust, which would affect word ratings and word recognition latencies. We also predicted that such person-based factors as disgust proneness and political ideology will mediate the effect. A word rating (Experiment 1) and a lexical decision (Experiment 2) study found partial support for these hypotheses. In brief, the main findings were as follows:

- More liberal participants rated the stimuli as more disgusting after being exposed to the headlines downplaying the threat of COVID-19, whereas more conservative participants gave higher disgust ratings following the headlines emphasizing it.

- More liberal participants were more extreme with their ratings and gave a broader range of responses (rating disgusting words as more disgusting and negative words as more negative, as well as rating non-disgusting words as less disgusting and positive words as more positive) than their more conservative peers regardless of the condition.

- Disgusting and negative words had a facilitatory effect for more liberal participants (shorter RTs) and an inhibitory effect for more conservative participants (longer RTs).

- More disgust-prone individuals rated everything as more disgusting than low disgust-prone ones.
- In the severe condition, low disgust-prone participants rated all the stimuli as more disgusting and negative, whereas high disgust-prone participants only rated low to moderately disgusting words as more disgusting and negative.

## Affective word ratings and political ideology

As expected, political orientation had a clear impact on word ratings. As we noted in the Introduction, the perception of the severity of the virus became an identity marker for both ends of the political spectrum. Given such drastic polarization, it is not surprising that the two types of the headlines produced the exact opposite effect on the participants depending on their political ideology. More liberal participants in our study were more disgusted by the headlines *downplaying* the severity of COVID-19 than those emphasizing it, rating everything as more disgusting afterwards. In contrast, more conservative participants assigned higher disgust ratings following the severe headlines. We offer two possible explanations for this result, one based on direct affective evaluation and the other on the disgust system response due to stimulus habituation. According to the first explanation, the stance on the virus from the political outgroup (downplaying headlines for the liberal participants and severe headlines for the conservative ones) evoked a strong emotional response, which translated into higher ratings on the disgust scale. However, in this case, one would also expect lower ratings on the valence scale depending on the headline, and this was not what we found. That leaves us the second possibility. As the liberal narrative revolves around the costs of not treating the virus seriously enough, they may have become habituated and desensitized to it. The take on the danger of COVID-19 may thus be perceived as the "default" by them and no longer alert their disgust system, whereas headlines contradicting this view might instantly elevate their disgust levels, signaling danger. The opposite, of course, should be true for more conservative participants. Note that one of our hypotheses was that more conservative participants will discard the severe headlines as alarmist, since previous research showed that conservatives have less trust in contradictory media and firmly believe that COVID-19 does not pose big health risks [55]. The current findings suggest that this was either not the case or, if it was the case, it did not stop their disgust system from ramping up. All in all, this is in line with mounting evidence that conservatives are more prone to disgust [49–51]. Even though previous studies found conservatives to be less concerned about the pandemic and less eager to engage in social distancing than liberals [55, 70], our results show that highlighting the danger of the virus still makes conservative participants give higher disgust ratings. Whether this translates into more adherence to safety protocols is a topic for further research.

One novel finding of our study is more extreme disgust and valence ratings by more liberal participants compared to their more conservative peers regardless of the condition. Disgusting and negative words were rated as more disgusting and more negative by more liberal participants, and the opposite was true for non-disgusting and positive words. We are not aware of any research examining whether political ideology correlates with ratings' extremity. It is entirely possible that this broader range of ratings is additionally mediated by some other personality traits and this needs to be verified by future research.

## Affective word ratings and disgust proneness

Disgust proneness affected word ratings over and above the effects of political ideology. Regardless of the headline type, more disgust-prone individuals rated all the stimuli as more

disgusting and negative stimuli as more negative than less disgust-prone individuals. This demonstrates that disgust ratings can serve as a good proxy for participant's disgust and adds to the growing body of evidence regarding the effects of disgust proneness on cognition in general and language processing in particular. [40, 41] found that disgust sensitivity was positively correlated with pupil dilation during the processing of stereotype-based clashing statements, suggesting that more disgust-prone individuals may experience greater arousal when interacting with stimuli that are disgusting either physically or morally. The results of the current study further indicate that even single word processing can be significantly affected by the participant's disgust sensitivity. In addition, disgust proneness significantly interacted with the headline type. While low disgust-prone participants rated all the stimuli as more disgusting and negative when the threat of COVID-19 was highlighted (severe headlines), high disgust-prone participants only rated low and moderately disgusting words as more disgusting and positive words as more negative in the severe condition. As we addressed in the Discussion after Experiment 1, this may be due to the ceiling effect since ratings assigned by high-disgust prone participants to extremely valenced stimuli were very close to the top of the disgust scale and the bottom of the valence scale.

## Political ideology vs disgust proneness in lexical access: The role of the pandemic

Our findings from the lexical decision experiment partially corroborated and extended the results for French by [46]. The authors found that disgusting words had a facilitatory effect for lexical recognition in less disgust-prone participants and an inhibiting effect in more disgust-prone participants. Our study, however, found that political ideology was *more predictive* of RTs than disgust sensitivity. Even though the general direction of the effect was the same (more liberal participants patterned like less disgust-prone ones), only political ideology significantly improved the model's fit when both factors were examined together. Overall, disgusting and negative words had a facilitatory effect on word recognition for more liberal participants and an inhibitory effect for more conservative participants. To the best of our knowledge, the interaction between political ideology and lexical decision times has not yet been researched. One possible explanation for the dominant effect of political ideology in our study is a big political component pertinent to the ongoing pandemic from its very beginning. [55] suggested that political ideology was uniquely predictive of the participant's COVID-19 behavior even when controlling for such variables as belief in science and COVID-related anxiety. Thus, it may be that political views have temporarily become a more salient marker of the behavioral immune system response than disgust sensitivity per se. This is, of course, a speculative idea that needs to be addressed by further research. One way to verify this would be to conduct the same study during the pandemic and post-pandemic.

## Differential effects of traits and states on lexical access

We did not find an effect of dynamically changing disgust levels (induced by headlines) on lexical access. Even though the headlines successfully affected participants' ratings in Experiment 1, they did not have an effect on RTs in Experiment 2—neither by themselves nor in interaction with person-based factors. Unlike headlines, however, political ideology *was* found predictive of word recognition latencies. Why would that be the case? Previous research has found political views to be just one manifestation of a cognitive and affective make-up and to have a robust correlation with threat perception [74]. It is thus not surprising that aligning with a particular political ideology may make disgust-related concepts in long-term memory more or less accessible (see [75] for converging findings with threat-related concepts). Thus,

our results suggest a difference between fluctuating states (i.e., the participant's emotional response to a particular set of headlines) and stable traits (i.e., aligning with more conservative or more liberal ideology) in affecting the ease of accessing disgusting and negative words. One alternative possibility to consider is that an exposure to COVID-related news may need to be longer to see an effect on lexical decision (we only showed a handful of headlines that the participants could switch through at their own pace). This could be tested by future research.

## Limitations of present research

Our study had several limitations that need to be noted. First and foremost, we did not collect participants' socioeconomic status, belief in science, self-perceived likelihood of contracting COVID-19, or COVID-19 related anxiety. Second, a convenience sample of university students produced a slightly skewed distribution of gender, age, and political ideology (most participants were young and more liberal females), which may have affected the results. That said, within the range of scores obtained in this experiment, the distribution was very close to normal.

One other concern needs to be addressed. Since our headlines reported on the pandemic, it is important to make sure that word recognition latencies were not affected by the presence of words directly related to the pandemic and to disease in general. As no lists of pandemic-related words exist, it is difficult to estimate how many words in the final dataset satisfied this criterion. Using our best judgment, we counted 6 out of 99 words that were disease-related, with the disgust indexes given in brackets: "unhealthy" (-0.7), "germ" (0.9), "sickening" (1.24), "deadly" (0.8), "parasite" (1.5), "disease" (1). Three of those words occurred in the severe headlines in full ("sickening", "disease", "deadly") and one in part ("bloodthirsty"–"blood"). As one can see, the words were relatively dispersed on the disgust scale. To make sure the results of Experiment 2 were not contaminated by this overlap, we reran the models without these four words. While disgust proneness was no longer significant, political ideology remained significant. This, once again, testifies to the stability of the effect of political ideology.

All in all, our studies found that not only do headlines about the pandemic affect the participants' disgust levels but that they also interact with a range of person-based factors, namely how prone the participant is to disgust and what political ideology they align with.

## Conclusion

The study shows that dynamic disgust levels affect word ratings but not the core language processing mechanisms, such as lexical access. It also provides tentative evidence that the politicization and polarization of the pandemic has led to tangible consequences in how strongly an individual's disgust system activates in response to different types of headlines.

## Supporting information

**S1 File. Contains all the supporting stimuli and tables.**
(DOCX)

## Acknowledgments

We thank the three reviewers and the editor whose insightful comments and suggestions have substantially improved this manuscript.

## Author Contributions

**Conceptualization:** Isabell Hubert Lyall, Juhani Järvikivi.

**Formal analysis:** Veranika Puhacheuskaya, Isabell Hubert Lyall.

**Methodology:** Veranika Puhacheuskaya, Isabell Hubert Lyall, Juhani Järvikivi.

**Software:** Veranika Puhacheuskaya.

**Supervision:** Juhani Järvikivi.

**Visualization:** Veranika Puhacheuskaya.

**Writing – original draft:** Veranika Puhacheuskaya.

**Writing – review & editing:** Veranika Puhacheuskaya, Juhani Järvikivi.

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
