## [Decision Letter · Decision Letter 0]

23 Nov 2021

PONE-D-21-29311COVIDisgust: Language Processing through the Lens of a Pandemic

PLOS ONE

Dear Dr. Puhacheuskaya,

Thank you for submitting your manuscript to PLOS ONE. After careful consideration, we feel that it has merit but does not fully meet PLOS ONE’s publication criteria as it currently stands. Therefore, we invite you to submit a revised version of the manuscript that addresses the points raised during the review process.

Your manuscript (PONE-D-21-29311) was read by three expert reviewers. Their comments are attached below. Reviewer 1 is Huili Wang. As an academic editor, I have read the manuscript myself. As you will see, all reviewers found some merits in your study. However, they also recommended that the manuscript should be greatly improved before it is published in PLOS ONE. Reviewer 1 suggested that the logic in the storyline be reconsidered. Reviewers 2 and 3 commented that more information and justification should be provided in the method and result sections. I largely agree with the reviewers, and I request that you respond to all their comments. 

The most serious issue, from my perspective, is the storyline. After the first two paragraphs, the hypothesis is presented rather abruptly in line 31. It is not crystal clear why you opted for the current research design. Please explain, for the purpose of studying the the effect of COVID-19-related news on people's disgust, why you asked participants to respond to words and used single word processing "as a proxy for disgust" (line 13). Please describe what the advantage of this procedure is. In the current introduction, it is also not clearly described why you opted for the word rating experiment and what the lexical decision task offers on top of the rating experiment. If your goal is to also study single word processing mechanism, then I would like you to describe more clearly what is missing in the previous research and what the present study offers. Although you are stating the importance of testing the plain text effect in line 84, this is no longer mentioned in the rest of the manuscript, and your stimuli unfortunately contained illustrations (line 116, see also Reviewer 3's comment). In addition, your conclusion is not supported by the data because you did not study "an individual's response to news about COVID-19" (line 362); what you studied was individuals' responses to words with a prior presentation of news about COVID-19 (see also Reviewer 1's comment).

In addition to reviewers' suggestions for the method and result sections, I am also concerned about your choice for the statical analysis. Although I agree that the GAMM can offer an interesting insight in many occasions, I am not fully convinced that the GAMM is the best choice in this study. Neither your predictions nor your interpretation of the results involves nonlinearity. If you choose to retain the GAMM analyses, please describe more clearly why it is important to consider nonlinearity for this topic. Otherwise, the three-way wiggly interactions look unnecessarily complex, and they might not attract a wide range of readers. For this reason, I strongly recommend that you also report (generalized) linear mixed-effects models. Assuming that the results are comparable between the GAMM and the LMM/GLMM, I prefer to see the LMM/GLMM in the main text and the GAMM in the supplementary material.

Finally, here are my line-by-line comments:

line 104: Please spell out "VAD."

line 175: the difference plot is crucial for readers to digest the three-way interaction fully. I request that the difference plot be presented together with Fig 2. This is applicable to all difference plots reported in this manuscript.

line 245: Please double-check whether you analyzed -1000/RT. Given the intercept and the slope in Table 4, as well as the values shown in Figure 5, I suspect that you analyzed -1/RT.

Figures: Please refrain from using different labels for the same variables. stand.p.disgust should be DS-R, and stand.p.politics should be W-P. 

In light of the reviewers' recommendation, my editorial decision is "Major Revision." If and only if you find it possible to satisfy the reviewers' and my requests, please revise and resubmit your manuscript. Please note that this does not guarantee eventual acceptance of your manuscript. If resubmitted, depending on the quality of the revision, I might send it to the same reviewers or reject it at the editorial stage. 

We look forward to receiving your revised manuscript.

Kind regards,

Koji Miwa, Ph.D.

Academic Editor

PLOS ONE

Journal Requirements:

"This research was supported by a Social Sciences and Humanities Research Council of 

Canada (http://www.sshrc-crsh.gc.ca/) Partnership Grant (Words in the World, 

895-2016-1008)."

"This research was supported by a Social Sciences and Humanities Research Council of Canada (http://www.sshrc-crsh.gc.ca/) Partnership Grant (Words in the World, 895-2016-1008). The funders had no role in study design, data collection and analysis, decision to publish, or preparation of the manuscript."

Reviewers' comments:

Reviewer's Responses to Questions

**Comments to the Author**

1. Is the manuscript technically sound, and do the data support the conclusions?

Reviewer #1: Partly

Reviewer #2: Yes

Reviewer #3: Yes

2. Has the statistical analysis been performed appropriately and rigorously? 

Reviewer #1: Yes

Reviewer #2: Yes

Reviewer #3: I Don't Know

3. Have the authors made all data underlying the findings in their manuscript fully available?

Reviewer #1: Yes

Reviewer #2: Yes

Reviewer #3: Yes

4. Is the manuscript presented in an intelligible fashion and written in standard English?

Reviewer #1: Yes

Reviewer #2: Yes

Reviewer #3: Yes

5. Review Comments to the Author

Reviewer #1: The paper titled “COVIDisgust: Language Processing through the Lens of a Pandemic” investigated whether the media’s stance on the COVID-19 pandemic can affect an individual’s disgust levels. The manuscript is technically sound as the experiments were carried out in a rigorous fashion with relevant variables being appropriately controlled. The statistical analysis was highly detailed and scientific with all data underlying the findings being fully available. In addition, the manuscript was written in standard English in an intelligible fashion. However, several issues should be addressed before the publication of the paper.

1. The research question of this paper is “whether the media’s stance on the COVID-19 pandemic can affect an individual’s disgust levels”, however, the conclusion drawn seemed to be the other way around, namely, an individual’s disgust levels would affect his or her response to news about the COVID-19 pandemic. The author should reconsider the conclusion drawn and clarify the logical connections between the hypotheses and results.

2. It would be better to present a brief summary of the experiment and the results in the first paragraph of the discussion section rather than introducing new information and questions.

3. Political ideology seemed to play an important role in manipulating participants’ responses to news about COVID-19 as the author stressed in the discussion and conclusion sections. It is recommended to incorporate it into the title and the research question.

4. The discussion about the lexical decision task could be enriched and extended with several citations.

5. One minor issue in the participants section, the author indicated that the participants’ self-reported English proficiency in the two experiments were 4.5 and 4.6 respectively. Could the author provide more details about how the score was calculated including questions or standards adopted?

Reviewer #2: This is an interesting study which examines the role of media information, disgust sensitivity and political orientation on word affective ratings and processing. Many studies have been conducted on the psychological consequences of the COVID pandemic. The main contribution of this work is that it focuses on a less explored issue, that is, the cognitive consequences of the COVID pandemic, in terms of language processing, relating them with individual differences. Clearly, the study is of interest to PlosOne readers. There are several issues, however, that need to be addressed before it is accepted for publication. I list them below:

Materials

-More information about the materials is needed. In particular:

1) Do the disgust-related words have high ratings only in disgust (and low ratings in other discrete emotions)?. That is, are they “pure” disgust words? (See Ferré et al., 2017, and Syssau et al., 2020, in Behavior Research Methods, for a distinction between pure and non-pure emotion related words).

2) Are high and low disgust words (with low valence) matched in arousal?

3) Are the three groups of words matched in the lexico-semantic variables that are known to affect language processing in general, and lexical decision tasks in particular?: Lexical frequency, length, age of acquisition, concreteness, cognate status of words (i.e., orthographic overlap between the English words and their translations in the native language of the bilingual participants).

4) Are there words directly related with pandemics among the data set? How many?

5) Please, include an appendix with the materials.

6) More information should be provided about the newspaper headlines. How many headlines were presented to the participants? It would be useful to have them in an appendix.

7) How were the pseudowords of Experiment 2 created?

Participants

-Almost half of the participants are not native speakers of English. There is much evidence of differences in emotional language processing in native and non-native languages. Part of them has been obtained with the lexical decision task, which is also used here (see, for instance, the work by Dewaele, Pavlenko, Caldwell-Harris, Costa, Ferré, Duñabeitia, etc). There is also evidence of differences in affective ratings between the L1 and the L2 (see, for instance, the work by Imbault, Vélez-Uribe or Prada). The authors should examine if there are differences in affective ratings (Experiment 1), as well as in emotional word processing (Experiment 2) between the native and non-native English participants.

Analyses

-In Experiment 2, the effect of the above mentioned variables (i.e., arousal, lexical frequency, length, age of acquisition, concreteness, cognate status) needs to be considered, as they are known to affect word processing in the lexical decision task. In particular, it is important to demonstrate that there is not a confounding effect of arousal, since high disgusting words seem to be more arousing than low disgusting words.

Discussion

-A discussion for each experiment should be included. In the General Discussion, the results of both experiments need to be integrated, trying to provide any explanation for the distinct results across experiments).

Minor

-A relevant reference is lacking: Silva et al. (2012), who explored the role of disgust sensitivity on the processing of disgust-related words in a lexical decision task.

Reviewer #3: This is an interesting article with a very thought-provoking topic related to COVID-19. The current research investigated how headlines related to COVID-19 influence on people’s word perception in terms of disgust level. In addition, the authors included participants’ political status as a possible influential factor in how they perceive words. Experiment 1 was to discover participants’ rating scale on how disgusting the participants feel towards words after viewing headlines. Experiment 2 was to discover the reaction time on a lexical decision task(LDT) after viewing headlines. Overall, they discovered the significant interaction between individual disgust sensitivity and political ideology. In short, liberals rated words as more disgusting after reading headlines compared to conservatives. In addition, they found that less conservative participants spent less RT on disgust words during LDT, which may be explained by the relation between the disgust level of words and long-term memory.

What follows is a page-by-page response to points in the article. The responses were divided into minor and major issues. The symbol '>>>' with page number(s) introduces a quote from the article, and is followed by my query.

<minor issues="">

p.2

>>> Prior linguistic research has shown that affective content of words influences how fast they are recognized [9, 10], which also interacts with a range of person-based factors, such as age [11–13], sex [14], character traits and mood [15, 16], native speaker status [17, 18] and others.

In this study, both native and non-native speakers were included as participants. Was there any difference between them in both Experiment 1 and 2 besides RT for the lexical decision task?

If there is any difference between them in terms of arousal and political status and their effect on RT in Experiment 2, please describe in detail. If no, just state that the difference between them was not observed in this research other than RT.

p.2

>>> Conservatives tend to score higher on disgust in general [25–27], and making people physically disgusted shifts their attitudes to the conservative end of the political spectrum [28].

p.6

>>> left-leaning but not right-leaning participants rated all positive words as more positive and negative words as more negative.

The authors use the words conservatives/liberals, and right-learning/left-learning interchangeably. They are synonyms, but conservatives/liberals sound more general whereas right-learning/left-learning sound more related to politics and policies. To add consistency, it would be recommended to stick to either of them.

p.3

>>> Mean self-reported English proficiency was 4.5

Out of what? Also, please add SD with the score.

p.4

>>>PsychoPy3

This should be properly cited as stated here (https://psychopy.org/about/index.html)

p.4

>>> DS-R

The abbreviation should be fully spelled when it appears for the first time.

p.4

>>> Likert-type scale

In this section, please describe each ratio more in detail. For example, the maximum score of the scale is unclear. To increase the replicability, let readers know how they can precisely replicate your experiments.

p.4

>>> In addition to not treating categorical data as continuous

Do you mean, not treating continuous data as categorical?

p.5

>>> if the participants were exposed to the Type I (severe) headlines.

I would recommend you to change the term Type I/II to avoid possible misinterpretation. Type I/II sounds more familiar to me in the statistical contexts.

p.5

>>> The best-fitting model for disgust ratings included the following significant predictors:

As GAM is a relatively new statistical approach in the linguistic field research, it would be appreciated if you could add some sentences to describe what the best-fitting model means and how you found the best-fitting model instead of just stating “Deviance explained = 30.5%.”.

p. 7

>>> Mean self-reported English proficiency was 4.6.

4.6 out of what? Also, please add SD with the score.

p.5, p.8

>>> A Student’s t-test

This is a tiny point, but I am not a big fan of the expression “student’s” (this is because this article is not so related to education that there is no need to treat participants as students). A participant’s t-test sound more natural.

<major issues="">

p.3

>>> It is therefore important to know whether plain texts (some of which also featured neutral, non-disgusting imagery) have the potential to produce the same effect as affective imagery.

p.4

>>> We then made screenshots of the headlines, some of which also had illustrations. No disturbing imagery was used.

From the sentence on p.3, I interpreted the authors were aiming to investigate whether plain texts about COVID-19 issues affect the feeling towards words as the imagery does. Yet, on p.4, some materials included illustrations. Therefore, what they wanted to achieve by making the difference from the previous research is unclear. In addition, although the authors states that disturbing imagery was not used, it is unclear how they distinguished the images whether they are disturbing or not.

p.7

>>> The procedure was the same as in Experiment 1 except for the main task and a short practice session before it.

Is it possible to add a figure to describe the procedure with images?

Each session seems to start with seeing the headline (s?) before the main task in both Exp1 and 2, but how often they perceived the headline(s) is not clearly described. My understanding is that the participants saw the headline 99 times in Exp 1 simply because there were 99 words to rate. But for LDT in Exp 2, for sure the authors had filler items so that how many times the participants saw the headline is unclear.

p.9

>>> However, prior studies have not explored how the feeling of disgust interacts with lexical decision times. It is thus possible that being more disgusted may in fact make the associated concepts in the long-term memory more accessible, facilitating recognition of highly disgusting words.

p.10

>>> This may indicate that being more disgusted may make disgust-related concepts in the long-term memory more accessible, facilitating recognition of associated words. More research on the topic would be valuable.

I see the points, but the discussion of long-term memory lacks a logical explanation at this moment. It is worth to state this is a new discovery of this article, which the previous research could not find. However, clear explanation why the results are showing the possibility of the relation with long-term memory should be clearly and precisely given with citations.</major></minor>

6. PLOS authors have the option to publish the peer review history of their article (what does this mean?). If published, this will include your full peer review and any attached files.

Reviewer #1: **Yes: **王慧莉

Reviewer #2: No

Reviewer #3: No

---

## [Author Response · Author response to Decision Letter 0]

9 Mar 2022

PONE-D-21-29311

COVIDisgust: Language Processing through the Lens of a Pandemic

PLOS ONE

Thank you for submitting your manuscript to PLOS ONE. After careful consideration, we feel that it has merit but does not fully meet PLOS ONE’s publication criteria as it currently stands. Therefore, we invite you to submit a revised version of the manuscript that addresses the points raised during the review process.

Your manuscript (PONE-D-21-29311) was read by three expert reviewers. Their comments are attached below. Reviewer 1 is Huili Wang. As an academic editor, I have read the manuscript myself. As you will see, all reviewers found some merits in your study. However, they also recommended that the manuscript should be greatly improved before it is published in PLOS ONE. Reviewer 1 suggested that the logic in the storyline be reconsidered. Reviewers 2 and 3 commented that more information and justification should be provided in the method and result sections. I largely agree with the reviewers, and I request that you respond to all their comments. 

The most serious issue, from my perspective, is the storyline. After the first two paragraphs, the hypothesis is presented rather abruptly in line 31. It is not crystal clear why you opted for the current research design. Please explain, for the purpose of studying the the effect of COVID-19-related news on people's disgust, why you asked participants to respond to words and used single word processing "as a proxy for disgust" (line 13). Please describe what the advantage of this procedure is. In the current introduction, it is also not clearly described why you opted for the word rating experiment and what the lexical decision task offers on top of the rating experiment. If your goal is to also study single word processing mechanism, then I would like you to describe more clearly what is missing in the previous research and what the present study offers. 

Author’s comment: Thank you for pointing this out. We have re-written the Introduction. We have clarified what was missing in previous research on affective word processing and how the present study corroborates and extends previous findings. We hope that our goals are now more transparent.

Although you are stating the importance of testing the plain text effect in line 84, this is no longer mentioned in the rest of the manuscript, and your stimuli unfortunately contained illustrations (line 116, see also Reviewer 3's comment).

Author’s comment: Unlike stimuli used in prior research, for instance those from the DIRTI database (Haberkamp A, Glombiewski JA, Schmidt F, Barke A. The DIsgust-RelaTed-Images (DIRTI) database: Validation of a novel standardized set of disgust pictures. Behaviour Research and Therapy. 2017 Feb 1;89:86–94.), all our illustrations were neutral and did not fall under an existing list of categories known to evoke disgust, such as bodily products, injuries/infections, deaths, etc. However, we decided to remove this paragraph from the main text due to that not being of primary importance for this study. Additionally, the full list of headlines can now be found in Appendix 2 for a review.

In addition, your conclusion is not supported by the data because you did not study "an individual's response to news about COVID-19" (line 362); what you studied was individuals' responses to words with a prior presentation of news about COVID-19 (see also Reviewer 1's comment).

Author’s comment: We believe that this is a reasonable generalization since disgust ratings act as a proxy for disgust and thus indicate how strongly an individual responded to a particular set of headlines about COVID-19. We have also rewritten the abstract, the discussion, and the conclusions so that the link between the participants’ response to news and their response to disgusting words is more obvious.

In addition to reviewers' suggestions for the method and result sections, I am also concerned about your choice for the statical analysis. Although I agree that the GAMM can offer an interesting insight in many occasions, I am not fully convinced that the GAMM is the best choice in this study. Neither your predictions nor your interpretation of the results involves nonlinearity. If you choose to retain the GAMM analyses, please describe more clearly why it is important to consider nonlinearity for this topic. Otherwise, the three-way wiggly interactions look unnecessarily complex, and they might not attract a wide range of readers. For this reason, I strongly recommend that you also report (generalized) linear mixed-effects models. Assuming that the results are comparable between the GAMM and the LMM/GLMM, I prefer to see the LMM/GLMM in the main text and the GAMM in the supplementary material.

Author’s comment: As per your suggestion, we are now reporting the results of LMM in the main text. Since the results of LMM and GAMM were virtually identical, we have decided to only provide the LMM analysis in the main text and to not add GAMM in Supplementary Material since it would not add anything to the paper. However, the full GAMM analysis with all the scripts and plots will be available on OSF in the main project folder (we have added this information to the main text).

Finally, here are my line-by-line comments

line 104: Please spell out "VAD." 

Author’s comment: We have spelled it out.

line 175: the difference plot is crucial for readers to digest the three-way interaction fully. I request that the difference plot be presented together with Fig 2. This is applicable to all difference plots reported in this manuscript.

Author’s comment: This comment is no longer applicable since we are reporting LMMs in the main text as per your request.

line 245: Please double-check whether you analyzed -1000/RT. Given the intercept and the slope in Table 4, as well as the values shown in Figure 5, I suspect that you analyzed -1/RT.

Author’s comment: We specified in the data analysis section that RTs were first reciprocally transformed (-1000/RTs) and then multiplied by 1000 to avoid extremely small numbers (and to make the plots look more intuitive, with shorter RTs being in the lower end of the y-axis). The range of reciprocally transformed RTs was -2247 to -433, which corresponds to 445 to 2305 ms of untransformed RTs.

Figures: Please refrain from using different labels for the same variables. stand.p.disgust should be DS-R, and stand.p.politics should be W-P. 

Author’s comment: Thank you for spotting this. We have changed the labels accordingly. 

In light of the reviewers' recommendation, my editorial decision is "Major Revision." If and only if you find it possible to satisfy the reviewers' and my requests, please revise and resubmit your manuscript. Please note that this does not guarantee eventual acceptance of your manuscript. If resubmitted, depending on the quality of the revision, I might send it to the same reviewers or reject it at the editorial stage. 

 

Reviewer #1: The paper titled “COVIDisgust: Language Processing through the Lens of a Pandemic” investigated whether the media’s stance on the COVID-19 pandemic can affect an individual’s disgust levels. The manuscript is technically sound as the experiments were carried out in a rigorous fashion with relevant variables being appropriately controlled. The statistical analysis was highly detailed and scientific with all data underlying the findings being fully available. In addition, the manuscript was written in standard English in an intelligible fashion. However, several issues should be addressed before the publication of the paper.

1. The research question of this paper is “whether the media’s stance on the COVID-19 pandemic can affect an individual’s disgust levels”, however, the conclusion drawn seemed to be the other way around, namely, an individual’s disgust levels would affect his or her response to news about the COVID-19 pandemic. The author should reconsider the conclusion drawn and clarify the logical connections between the hypotheses and results.

Author’s comment: While it is true that the participant’s disgust level itself also affected their word ratings and lexical decisions times, this was not the main result of the study. The main result of the experiment showed that our experimental manipulation, type of the headline (“the media’s stance”), significantly modulated participants' word ratings and, additionally, interacted with individual disgust proneness and political ideology. As the suggested change would not capture this result in its entirety or the direction of the observed effects, we therefore have no empirical grounds to change the conclusion. 

2. It would be better to present a brief summary of the experiment and the results in the first paragraph of the discussion section rather than introducing new information and questions.

Author’s comment: Thank you for this suggestion. We have rewritten the discussion accordingly.

3. Political ideology seemed to play an important role in manipulating participants’ responses to news about COVID-19 as the author stressed in the discussion and conclusion sections. It is recommended to incorporate it into the title and the research question.

Author’s comment: We have changed the title.

4. The discussion about the lexical decision task could be enriched and extended with several citations.

Author’s comment: We have described our predictions, hypotheses, and discussion of the results for Experiment 2 in more detail.

5. One minor issue in the participants section, the author indicated that the participants’ self-reported English proficiency in the two experiments were 4.5 and 4.6 respectively. Could the author provide more details about how the score was calculated including questions or standards adopted?

Author’s comment: We have added a clarification to the Participants section. The participants were simply asked to rate their English proficiency on a 5-point scale. No tests were used.

 

Reviewer #2: This is an interesting study which examines the role of media information, disgust sensitivity and political orientation on word affective ratings and processing. Many studies have been conducted on the psychological consequences of the COVID pandemic. The main contribution of this work is that it focuses on a less explored issue, that is, the cognitive consequences of the COVID pandemic, in terms of language processing, relating them with individual differences. Clearly, the study is of interest to PlosOne readers. There are several issues, however, that need to be addressed before it is accepted for publication. I list them below:

Materials

-More information about the materials is needed. In particular:

1) Do the disgust-related words have high ratings only in disgust (and low ratings in other discrete emotions)?. That is, are they “pure” disgust words? (See Ferré et al., 2017, and Syssau et al., 2020, in Behavior Research Methods, for a distinction between pure and non-pure emotion related words).

Author’s comment: We did not control for other discrete emotions in this study, and our words were not “pure” disgust words since the goal of the article was to examine how the word’s disgust score interacts with other variables rather than to examine the differential contribution of discrete emotions. Ferré et al. (2017) actually did not find any differences in RTs between disgusting and fearful words in any of their experiments (albeit their stimuli were Spanish, not English).

We checked the NRC Emotion Intensity Lexicon for other emotions post-factum and it turned out that a half or more of our 33 high disgust words did not have associated anger, fear and sadness scores, and only 2 had anticipation scores. Mean fear score associated with the remaining words was 0.72, mean anger score 0.62, mean sadness score 0.73. Since there are so many missing scores, it is not possible to add the scores from other emotions in our data analysis at this point. We also don’t think that keeping the fear score low for highly disgusting words is reasonable (or even manageable) because this will likely constitute a confound of its own, as highly disgusting words may naturally evoke fear due to their relatedness to disease. In fact, only 52 (due to word forms being present, even fewer) words in the NRC database had a disgust score > 0.5 and a fear score < 0.5 (range 0-1), with mean disgust score of 0.62 and mean fear score of 0.4. As you can see, these are not high disgust low fear words – these are mixed words with moderate fear and disgust scores. Further, at least one third of those words had extremely low lexical frequency (freq range = 8 - 100, log freq range = 1.6 to 2.6). To compare, the lowest lexical frequency of our stimuli was 211 (log 5.4). Thus, by getting rid of one potential confound we would decrease the power of the experiment due to only having mid- and low-disgust words, in addition to adding other confounds (frequency, prevalent religious and morality themes, etc.). 

2) Are high and low disgust words (with low valence) matched in arousal?

Author’s comment: When preparing the stimuli, we kept arousal values constant (between -1 and 1 in a normalized distribution). However, since the correlation between word disgust and arousal was still significant albeit small (r = 0.2, p < .04), we added arousal as a predictor in all our models (it turned out to be only marginally significant in 1 out of 6 models). We also added a table with correlation coefficients between all words’ characteristics to the main text.

3) Are the three groups of words matched in the lexico-semantic variables that are known to affect language processing in general, and lexical decision tasks in particular?: Lexical frequency, length, age of acquisition, concreteness, cognate status of words (i.e., orthographic overlap between the English words and their translations in the native language of the bilingual participants).

Author’s comment: Thank you for pointing this out. We have added all those variables (except for cognate status) in our models for the lexical decision experiment but not for the word rating experiment, since we don’t believe they have any effect on word ratings for disgust and valence. At this stage, it is unfortunately impossible to test the effects of cognate status since some participants chose the option “other” as their native language in the list of languages. Only age of acquisition was significant, with the predicted direction of its effect: words acquired earlier were reacted to faster. Adding these control variables did not change the overall results and conclusions of Experiment 2.

4) Are there words directly related with pandemics among the data set? How many?

Author’s comment: It is unfortunately very difficult to estimate it since no lists of pandemic-related words exist (to our knowledge). Using our best judgment, we counted 6 out of 99 words that could be more or less related to the pandemic (but also to disease in general, not specific to the current pandemic): unhealthy, germ, sickening, deadly, parasite, disease. Three words actually occurred in the severe headlines in full (“sickening”, “disease”, “deadly”) and one in part (“bloodthirsty” – “blood”). To make sure the results of Experiment 2 were not contaminated by this overlap, we additionally reran the models without these four words, and the main result did not change.

5) Please, include an appendix with the materials.

Author’s comment: We have added Appendix 1 with the stimuli. 

6) More information should be provided about the newspaper headlines. How many headlines were presented to the participants? It would be useful to have them in an appendix.

Author’s comment: We provided that information in the Materials section: “For the headline manipulation, we randomly selected seven news articles emphasizing the severity of COVID-19 and eight articles downplaying it.” We have now added the headlines in Appendix 2.

7) How were the pseudowords of Experiment 2 created?

Author’s comment: Pseudowords were created by modifying one or more letters in existing English words. A native speaker of English made sure they did not contain any real but archaic words, words that look like they could be a typo in real words, slang words, and words that do not currently exist but sound like a neologism. We have added this description into the main text. The full list of pseudowords is available in Appendix 1.

Participants

-Almost half of the participants are not native speakers of English. There is much evidence of differences in emotional language processing in native and non-native languages. Part of them has been obtained with the lexical decision task, which is also used here (see, for instance, the work by Dewaele, Pavlenko, Caldwell-Harris, Costa, Ferré, Duñabeitia, etc). There is also evidence of differences in affective ratings between the L1 and the L2 (see, for instance, the work by Imbault, Vélez-Uribe or Prada). The authors should examine if there are differences in affective ratings (Experiment 1), as well as in emotional word processing (Experiment 2) between the native and non-native English participants.

Author’s comment: Thank you for the suggestion. We have reclassified all participants who did not choose English as the first language they acquired in childhood as non-native speakers (rather than only those who did not choose English as their primary language) and aggregated self-reported English proficiency and SD by these two groups (native and non-native speakers) in the description of the participants. We are also now providing the results of fully specified models for both experiments, which show that native speaker status was a significant predictor only in Experiment 2 (lexical decision) but not in Experiment 1 (ratings). Moreover, rerunning the models for Experiment 2 with native speakers only did not change the main result. 

Analyses

-In Experiment 2, the effect of the above mentioned variables (i.e., arousal, lexical frequency, length, age of acquisition, concreteness, cognate status) needs to be considered, as they are known to affect word processing in the lexical decision task. In particular, it is important to demonstrate that there is not a confounding effect of arousal, since high disgusting words seem to be more arousing than low disgusting words.

Author’s comment: In addition to lexical frequency that was already in the models, we have now added word length, arousal, age of acquisition, and concreteness and report the results of the full models. At this stage, it is unfortunately impossible to test the effects of cognate status since some participants chose the option “other” as their native language in the list of languages. Only age of acquisition was significant. 

Discussion

-A discussion for each experiment should be included. In the General Discussion, the results of both experiments need to be integrated, trying to provide any explanation for the distinct results across experiments).

Author’s comment: We have added a discussion after each experiment and a general discussion in the end.

Minor

-A relevant reference is lacking: Silva et al. (2012), who explored the role of disgust sensitivity on the processing of disgust-related words in a lexical decision task.

Author’s comment: Thank you! We have incorporated the study into our predictions and discussion.

 

Reviewer #3: This is an interesting article with a very thought-provoking topic related to COVID-19. The current research investigated how headlines related to COVID-19 influence on people’s word perception in terms of disgust level. In addition, the authors included participants’ political status as a possible influential factor in how they perceive words. Experiment 1 was to discover participants’ rating scale on how disgusting the participants feel towards words after viewing headlines. Experiment 2 was to discover the reaction time on a lexical decision task(LDT) after viewing headlines. Overall, they discovered the significant interaction between individual disgust sensitivity and political ideology. In short, liberals rated words as more disgusting after reading headlines compared to conservatives. In addition, they found that less conservative participants spent less RT on disgust words during LDT, which may be explained by the relation between the disgust level of words and long-term memory.

What follows is a page-by-page response to points in the article. The responses were divided into minor and major issues. The symbol '>>>' with page number(s) introduces a quote from the article, and is followed by my query.

p.2

>>> Prior linguistic research has shown that affective content of words influences how fast they are recognized [9, 10], which also interacts with a range of person-based factors, such as age [11–13], sex [14], character traits and mood [15, 16], native speaker status [17, 18] and others.

In this study, both native and non-native speakers were included as participants. Was there any difference between them in both Experiment 1 and 2 besides RT for the lexical decision task?

If there is any difference between them in terms of arousal and political status and their effect on RT in Experiment 2, please describe in detail. If no, just state that the difference between them was not observed in this research other than RT.

Author’s comment: We are now providing the output of fully specified models for both experiments, which shows that the native speaker status was a significant predictor only in Experiment 2 but not in Experiment 1. We have also added a note into the Results section for Experiment 2 that excluding non-native speakers from the analysis did not change the main findings. Please also see our response to the comment about English proficiency below.

p.2

>>> Conservatives tend to score higher on disgust in general [25–27], and making people physically disgusted shifts their attitudes to the conservative end of the political spectrum [28].

p.6

>>> left-leaning but not right-leaning participants rated all positive words as more positive and negative words as more negative.

The authors use the words conservatives/liberals, and right-learning/left-learning interchangeably. They are synonyms, but conservatives/liberals sound more general whereas right-learning/left-learning sound more related to politics and policies. To add consistency, it would be recommended to stick to either of them.

Author’s comment: Thank you for this suggestion! Since we used the Wilson-Patterson’s Conservatism Scale for collecting political ideology, we decided to stick with conservatives/liberals and edited the article appropriately.

p.3

>>> Mean self-reported English proficiency was 4.5

Out of what? Also, please add SD with the score.

Author’s comment: We have specified that the proficiency was measured on a 5-point scale. We also reclassified all participants who did not choose English as the first language they acquired in childhood as non-native speakers (rather than only those who did not choose English as their primary language) and aggregated self-reported English proficiency and SD by these two groups (native and non-native speakers). We are now providing the results of fully specified models for both experiments, which show that native speaker status was a significant predictor only in Experiment 2 but not in Experiment 1. 

p.4

>>>PsychoPy3

This should be properly cited as stated here (https://psychopy.org/about/index.html)

Author’s comment: Thank you! We have added a proper citation.

p.4

>>> DS-R

The abbreviation should be fully spelled when it appears for the first time.

Author’s comment: We have spelled the abbreviation.

p.4

>>> Likert-type scale

In this section, please describe each ratio more in detail. For example, the maximum score of the scale is unclear. To increase the replicability, let readers know how they can precisely replicate your experiments.

Author’s comment: The scale is provided in the previous paragraph Procedure: “After that, the participants rated how disgusting a word feels to them (1 = “not at all” to 5 “extremely”) and how positive/negative it feels (1 = “very negative” to 5 “very positive”).

p.4

>>> In addition to not treating categorical data as continuous

Do you mean, not treating continuous data as categorical?

Author’s comment: Ratings elicited on a Likert scale yield discrete/categorical (ordinal) data since there is no strictly defined distance between the values of the scale, they can vary in magnitude between respondents, and they are ordered. Standard regression analyses treat such data as continuous. The advantage of GAMMs for ordinal data is precisely that they do not treat categorical data as continuous.

p.5

>>> if the participants were exposed to the Type I (severe) headlines.

I would recommend you to change the term Type I/II to avoid possible misinterpretation. Type I/II sounds more familiar to me in the statistical contexts.

Author’s comment: Thank you. We have removed the Type I/II terminology and now refer to them as “severe headlines” and “downplaying headlines”.

p.5

>>> The best-fitting model for disgust ratings included the following significant predictors:

As GAM is a relatively new statistical approach in the linguistic field research, it would be appreciated if you could add some sentences to describe what the best-fitting model means and how you found the best-fitting model instead of just stating “Deviance explained = 30.5%.”.

Author’s comment: The best-fitting model in our paper was obtained by doing stepwise forward selection (adding predictors one by one and checking whether an N+1 model produced a significant improvement over an N model). The compareML() function that we used for model comparison outputs a chi-square test of REML scores and an AIC difference between two models. If the chi-square test had a p-value of > .05, suggesting a non-significant difference in REML scores between a less and a more complex model, then the simpler model was preferred and the predictor was removed. Thus, the best-fitting model in our analysis was a maximally specified model that yielded a significant improvement over a simpler model. 

However, since it was requested that we reanalyze the data using linear mixed-effect modeling, this description is no longer relevant and will not be added to the main text.

p. 7

>>> Mean self-reported English proficiency was 4.6.

4.6 out of what? Also, please add SD with the score.

Author’s comment: We have specified that proficiency was measured on a 5-point scale. We have also added means and SDs for English proficiency for the two groups (native and non-native speakers).

p.5, p.8

>>> A Student’s t-test

This is a tiny point, but I am not a big fan of the expression “student’s” (this is because this article is not so related to education that there is no need to treat participants as students). A participant’s t-test sound more natural.

Author’s comment: Student’s t-test got its name from statistician William Sealy Gosset who published under the pen name Student. It is not related to the area of our paper. 

p.3

>>> It is therefore important to know whether plain texts (some of which also featured neutral, non-disgusting imagery) have the potential to produce the same effect as affective imagery.

p.4

>>> We then made screenshots of the headlines, some of which also had illustrations. No disturbing imagery was used.

From the sentence on p.3, I interpreted the authors were aiming to investigate whether plain texts about COVID-19 issues affect the feeling towards words as the imagery does. Yet, on p.4, some materials included illustrations. Therefore, what they wanted to achieve by making the difference from the previous research is unclear. In addition, although the authors states that disturbing imagery was not used, it is unclear how they distinguished the images whether they are disturbing or not.

Author’s comment: Although we explicitly stated that some of our headlines contained illustrations and that, unlike stimuli used in prior research such as those from DIRTI https://www.sciencedirect.com/science/article/abs/pii/S0005796716301978, all our illustrations were neutral (i.e., did not fall under categories known to evoke strong disgust, such as bodily products, injuries/infections, deaths, etc.), we have removed this paragraph from the main text due to that not being of primary importance for this study. Additionally, the full list of headlines can now be found in Appendix 2 for a review. 

p.7

>>> The procedure was the same as in Experiment 1 except for the main task and a short practice session before it.

Is it possible to add a figure to describe the procedure with images?

Each session seems to start with seeing the headline (s?) before the main task in both Exp1 and 2, but how often they perceived the headline(s) is not clearly described. My understanding is that the participants saw the headline 99 times in Exp 1 simply because there were 99 words to rate. But for LDT in Exp 2, for sure the authors had filler items so that how many times the participants saw the headline is unclear.

Author’s comment: We have edited the description of the procedure for clarity. The participants only saw a set of headlines once, in the beginning of each session before the main task started. They did not see the headlines before every trial.

p.9

>>> However, prior studies have not explored how the feeling of disgust interacts with lexical decision times. It is thus possible that being more disgusted may in fact make the associated concepts in the long-term memory more accessible, facilitating recognition of highly disgusting words.

p.10

>>> This may indicate that being more disgusted may make disgust-related concepts in the long-term memory more accessible, facilitating recognition of associated words. More research on the topic would be valuable.

I see the points, but the discussion of long-term memory lacks a logical explanation at this moment. It is worth to state this is a new discovery of this article, which the previous research could not find. However, clear explanation why the results are showing the possibility of the relation with long-term memory should be clearly and precisely given with citations.

Author’s comment: We have extended the discussion and added relevant citations.

---

## [Decision Letter · Decision Letter 1]

2 May 2022

PONE-D-21-29311R1COVIDisgust: Language Processing through the Lens of Partisanship

PLOS ONE

Dear Dr. Puhacheuskaya,

Thank you for submitting your manuscript to PLOS ONE. After careful consideration, we feel that it has merit but does not fully meet PLOS ONE’s publication criteria as it currently stands. Therefore, we invite you to submit a revised version of the manuscript that addresses the points raised during the review process.

Kudos for the effort and the time you devoted for the respectable revisions! As you can see all three reviewers responded to your comments and revisions more positively. I agree with the reviewers. The manuscript was improved in many respects. Although, as Reviewer 3 commented, it might be still challenging for many readers to digest the whole results with many interactions, I think you have done your job. Now, I will be happy if you can respond to the comments from Reviewers 1 and 3. In addition to the reviewers' comments, I have several comments of my own.

(1) Your motivation for the GAMM

You stated your motivation for the GAMM analysis as "Since ratings on a Likert-type scale yield ordinal data and thus, in principle, should not be analyzed with Gaussian family models assuming continuous data, we additionally ran a generalized additive mixed-modeling analysis (GAMM) for ordinal data" (p. 12). However, given this motivation, it might sound puzzling to the readers why you are not analyzing the data with something like the cumulative-link mixed-effects model for ordinal data (with the R package "ordinal"). I am not asking you to redo the whole analyses, but I would like you to reconsider your motivation for the GAMM. Doesn't your motivation have something to do with potential nonlinearity?

(2) RTs

The way you transformed RTs (after the reciprocal transformation) is not common, I personally think. Given that the resulting RTs are still different from the original RTs, not all readers might agree with your statement that "the plots can be read intuitively" (p. 24). However, I am not asking you to redo the analyses because it is not wrong either. I would just like you to explore optimal reporting method in the future. If an intuitive interpretation is what you want to achieve, I think a back-transformation can be applied when plotting the model-predicted values.

> RT = c(600, 700, 800) # for these sample RTs

> (-1000/RT)*1000 # this was done in this study

  # [1] -1666.667 -1428.571 -1250.000

> -1000/(-1000/RT) # back-transformation for the reciprocal transformation

  # [1] 600 700 800

(3) For all p-values and rs, please remove the zero before the decimal point.

(4) This might be my problem, but it seems that the interpretation for the W-P scale is not provided before the text "More liberal participants rated the stimuli as more disgusting..." (p. 15) It is therefore not clear, for "liberal," which part of the scale the reader should focus on.

(5) "The lines for the five levels of word disgust are..." (p. 16) might be misleading because it is not a factor with five levels, isn't it? Do you mean quantiles?

(5) Please double-check your statement on page 19: "The interaction between the participant's political ideology and valence ratings." Do you mean an interaction with word valence?

We look forward to receiving your revised manuscript.

Kind regards,

Koji Miwa, Ph.D.

Academic Editor

PLOS ONE

Journal Requirements:

Reviewers' comments:

Reviewer's Responses to Questions

**Comments to the Author**

1. If the authors have adequately addressed your comments raised in a previous round of review and you feel that this manuscript is now acceptable for publication, you may indicate that here to bypass the “Comments to the Author” section, enter your conflict of interest statement in the “Confidential to Editor” section, and submit your "Accept" recommendation.

Reviewer #1: All comments have been addressed

Reviewer #2: All comments have been addressed

Reviewer #3: All comments have been addressed

2. Is the manuscript technically sound, and do the data support the conclusions?

Reviewer #1: Yes

Reviewer #2: Yes

Reviewer #3: Partly

3. Has the statistical analysis been performed appropriately and rigorously? 

Reviewer #1: Yes

Reviewer #2: Yes

Reviewer #3: Yes

4. Have the authors made all data underlying the findings in their manuscript fully available?

Reviewer #1: Yes

Reviewer #2: Yes

Reviewer #3: Yes

5. Is the manuscript presented in an intelligible fashion and written in standard English?

Reviewer #1: Yes

Reviewer #2: Yes

Reviewer #3: Yes

6. Review Comments to the Author

Reviewer #1: Generally speaking, the authors revised as the reviewers suggested. However, there are some minor points that need to be explained or revised.

1.Line 14-16: In the abstract, I was wondering if the detailed reference information should be given in the parenthesis. My suggestion is to remove it.

2.Line 25: “after the headlines emphasizing it” seems ungrammatical. Please confirm.

3.In the section of “Introduction” and “Discussion”, I suggest adding some subtitles so that the ideas are more clearly elaborated. Furthermore, the ideas expressed in these two sections should be more symmetrical, being more focused on the hypotheses of the research.

Reviewer #2: (No Response)

Reviewer #3: Comments to the authors

The manuscript was well revised including additional explanations for materials and procedures as reviewers requested. The added explanations of the data analysis on page 11 sound convincing, and I respect how the authors changed the statistical analysis during the revision.

However, the manuscript is still challenging for readers to understand, and it probably requires readers to read several times to fully comprehend what the authors are trying to do.

Comments with (>>) below are some minor revisions and I hope that they could enhance the readers' comprehensibility of the manuscript.

p.7 line 134

untrustworthy, and be discarded by more conservative participants, resulting in no effect.

>>The authors could specify to which valuable (disgust rating?) the effect is going to be less influential.

p.28 line 515

Importantly, we did not find an effect of the headline on word recognition latencies. This suggests that fluctuating levels of disgust may not affect such core language processing mechanisms as lexical access. It may also mean that stable traits are more predictive of lexical recognition latencies than fluctuating states.

>>This explanation of the results of experiment 2 (LDT) is convincing. However, the result of no effect of COVID headlines makes readers confused a little bit.

Political ideology has an effect on online language (disgust words) processing, not limited to pandemic associated topics, is that what the authors are trying to say?

Probably additional sentences are needed to help readers understand the interpretation of the results and how the results can be generalized.

7. PLOS authors have the option to publish the peer review history of their article (what does this mean?). If published, this will include your full peer review and any attached files.

Reviewer #1: **Yes: **Wang Huili

Reviewer #2: No

Reviewer #3: No

---

## [Author Response · Author response to Decision Letter 1]

2 Jun 2022

Response to the Editor’s:

(1) Your motivation for the GAMM

You stated your motivation for the GAMM analysis as "Since ratings on a Likert-type scale yield ordinal data and thus, in principle, should not be analyzed with Gaussian family models assuming continuous data, we additionally ran a generalized additive mixed-modeling analysis (GAMM) for ordinal data" (p. 12). However, given this motivation, it might sound puzzling to the readers why you are not analyzing the data with something like the cumulative-link mixed-effects model for ordinal data (with the R package "ordinal"). I am not asking you to redo the whole analyses, but I would like you to reconsider your motivation for the GAMM. Doesn't your motivation have something to do with potential nonlinearity?

We have changed the text of the manuscript to “Initially the data was analyzed using generalized additive mixed modeling (GAMM) for ordinal data following (60). However, the resulting analysis was highly complex, and it was advised that we use linear mixed modeling instead. The results of the two analyses were virtually identical, and we will thus report linear mixed modeling for simplicity. The full GAMM analysis with the scripts and plots is available for review on OSF.” We hope this wording will be less confusing.

(2) RTs

The way you transformed RTs (after the reciprocal transformation) is not common, I personally think. Given that the resulting RTs are still different from the original RTs, not all readers might agree with your statement that "the plots can be read intuitively" (p. 24). However, I am not asking you to redo the analyses because it is not wrong either. I would just like you to explore optimal reporting method in the future. If an intuitive interpretation is what you want to achieve, I think a back-transformation can be applied when plotting the model-predicted values.

> RT = c(600, 700, 800) # for these sample RTs

> (-1000/RT)*1000 # this was done in this study

 # [1] -1666.667 -1428.571 -1250.000

> -1000/(-1000/RT) # back-transformation for the reciprocal transformation

 # [1] 600 700 800

Thank you! We will certainly keep it in mind for future analyses. To clarify, the comment about intuitive reading meant that, after this transformation, shorter RTs are still plotted in the bottom of the Y-axis and longer RTs at the top of the Y-axis, but we do see your point here.

(3) For all p-values and rs, please remove the zero before the decimal point.

We have removed the zeroes.

(4) This might be my problem, but it seems that the interpretation for the W-P scale is not provided before the text "More liberal participants rated the stimuli as more disgusting..." (p. 15) It is therefore not clear, for "liberal," which part of the scale the reader should focus on.

We have added the following clarification: “More liberal participants (lower W-P score)...”

(5) "The lines for the five levels of word disgust are..." (p. 16) might be misleading because it is not a factor with five levels, isn't it? Do you mean quantiles?

Thank you, we have corrected the error.

(5) Please double-check your statement on page 19: "The interaction between the participant's political ideology and valence ratings." Do you mean an interaction with word valence?

Thank you for spotting this, we have corrected this error.

 

Response to Reviewer 1:

Reviewer #1: Generally speaking, the authors revised as the reviewers suggested. However, there are some minor points that need to be explained or revised.

1.Line 14-16: In the abstract, I was wondering if the detailed reference information should be given in the parenthesis. My suggestion is to remove it.

We have removed the detailed reference. 

2.Line 25: “after the headlines emphasizing it” seems ungrammatical. Please confirm.

We have changed the sentences to “More liberal participants assigned higher disgust ratings after the headlines discounted the threat of COVID-19, whereas more conservative participants did so after the headlines emphasized it.” for more clarity.

3.In the section of “Introduction” and “Discussion”, I suggest adding some subtitles so that the ideas are more clearly elaborated. Furthermore, the ideas expressed in these two sections should be more symmetrical, being more focused on the hypotheses of the research.

We have added subtitles and restructured the Introduction and the General Discussion for more clarity.

 

Response to Reviewer 3:

Reviewer #3: Comments to the authors

The manuscript was well revised including additional explanations for materials and procedures as reviewers requested. The added explanations of the data analysis on page 11 sound convincing, and I respect how the authors changed the statistical analysis during the revision.

However, the manuscript is still challenging for readers to understand, and it probably requires readers to read several times to fully comprehend what the authors are trying to do.

Comments with (>>) below are some minor revisions and I hope that they could enhance the readers' comprehensibility of the manuscript.

p.7 line 134

untrustworthy, and be discarded by more conservative participants, resulting in no effect.

>>The authors could specify to which valuable (disgust rating?) the effect is going to be less influential.

We clarified the sentence: “Headlines emphasizing the threat will be considered uncredible, untrustworthy, and be discarded by more conservative participants, resulting in no effect in disgust ratings.”

p.28 line 515

Importantly, we did not find an effect of the headline on word recognition latencies. This suggests that fluctuating levels of disgust may not affect such core language processing mechanisms as lexical access. It may also mean that stable traits are more predictive of lexical recognition latencies than fluctuating states.

>>This explanation of the results of experiment 2 (LDT) is convincing. However, the result of no effect of COVID headlines makes readers confused a little bit.

Political ideology has an effect on online language (disgust words) processing, not limited to pandemic associated topics, is that what the authors are trying to say?

Probably additional sentences are needed to help readers understand the interpretation of the results and how the results can be generalized.

We have rewritten the paragraph and added more details to both the Discussion after Exp 2 and the General Discussion. Specifically, we have added a subsection “Differential effects of traits and states on lexical access” to General Discussion.

---

## [Editor Report · Decision Letter 2]

27 Jun 2022

COVIDisgust: Language Processing through the Lens of Partisanship

PONE-D-21-29311R2

Dear Dr. Puhacheuskaya,

We’re pleased to inform you that your manuscript has been judged scientifically suitable for publication and will be formally accepted for publication once it meets all outstanding technical requirements.

Kind regards,

Koji Miwa, Ph.D.

Academic Editor

PLOS ONE
---

## [Editor Report · Acceptance letter]

1 Jul 2022

PONE-D-21-29311R2 

COVIDisgust: Language processing through the lens of partisanship 

Dear Dr. Puhacheuskaya:

I'm pleased to inform you that your manuscript has been deemed suitable for publication in PLOS ONE. Congratulations! Your manuscript is now with our production department. 

Kind regards, 

on behalf of

Dr. Koji Miwa 

Academic Editor

PLOS ONE